# Model-Free Adversarial Purification via Coarse-To-Fine Tensor Network Representation

## Abstract

Deep neural networks are known to be vulnerable to well-designed adversarial attacks. Although numerous defense strategies have been proposed, many are tailored to specific attacks or tasks and often fail to generalize across diverse scenarios. In this paper, we propose Tensor Network Purification (TNP), a novel model-free optimization-based purification framework built upon a specially designed tensor network decomposition algorithm. TNP depends neither on the pre-trained generative model nor the specific dataset, resulting in robust generalization across diverse adversarial scenarios. To this end, the key challenge lies in relaxing Gaussian-noise assumptions of classical decompositions and accommodating the unknown distribution of adversarial perturbations. Unlike the low-rank representation of classical decompositions, TNP aims to reconstruct the unobserved clean example from an adversarial example. Specifically, TNP leverages progressive downsampling and introduces a novel adversarial optimization objective to address the challenge of minimizing reconstruction error but without inadvertently restoring adversarial perturbations. Extensive experiments conducted on CIFAR-10, CIFAR-100, and ImageNet demonstrate that our method generalizes effectively across various norm threats, attack types, and tasks, providing a versatile and promising adversarial purification technique.

## 1   Introduction

Deep neural networks (DNNs) have achieved remarkable success across a wide range of tasks. However, DNNs have been shown to be vulnerable to adversarial examples (Szegedy et al., 2014; Goodfellow et al., 2015), which are generated by adding small, human-imperceptible perturbations to natural images but completely incorrect the prediction results to DNNs with potentially disastrous consequences. This inherent vulnerability of DNNs underscores the critical need for robust defense mechanisms to mitigate adversarial attacks effectively.

Since then, numerous methods have been proposed to defend against adversarial examples. Notably, adversarial training (AT, Goodfellow et al., 2015) typically aims to retrain DNNs using specific adversarial examples, achieving robustness to seen types of adversarial attacks but performing poorly against unseen perturbations (Laidlaw et al., 2021). Another class of defense methods is adversarial purification (AP, Yoon et al., 2021), which leverages pre-trained generative models to remove adversarial perturbations and demonstrates better generalization than AT against unseen attacks (Nie et al., 2022; Lin et al., 2024a). However, AP methods heavily rely on pre-trained models tailored to specific datasets, limiting their transferability to different data distributions and tasks. As a result, both mainstream techniques face generalization challenges: AT struggles with diverse norm threats, and AP with task generalization, restricting their deployment to broader scenarios.

Submitted to 39th Conference on Neural Information Processing Systems (NeurIPS 2025). Do not distribute.

To address these challenges, we propose a novel model-free optimization-based adversarial purification framework built upon a coarse-to-fine tensor network decomposition, termed Tensor Network Purification (TNP), which bridges the gap between low-rank tensor network representation with Gaussian noise assumption and removal of adversarial perturbations with unknown distributions. As a model-free optimization-based technique, tensor network (TN) depends neither on any pre-trained generative model nor specific dataset (Oseledets, 2011; Zhao et al., 2016), enabling it to achieve strong generalization across diverse adversarial scenarios. As a pre-processing step, TN can eliminate potential adversarial perturbations for both clean and adversarial examples before feeding them into the classifier (Yoon et al., 2021), which also implies that TN can defend against adversarial attacks without retraining the classifier model. Moreover, by acting directly on a single input without fixed model parameters, TN is inherently more resistant to adversarial attacks, as discussed further in Appendix C. Consequently, benefiting from the aforementioned advantages, it is evident that TN-based adversarial purification represents a highly promising direction, offering the transferability to be effectively applied across diverse adversarial scenarios.

The existing TN methods are particularly favorable for image completion and denoising when the corruption is sparse or follows a Gaussian distribution as long as it can be modeled explicitly. However, the distribution of well-designed adversarial perturbations fundamentally differs from these assumptions and often aligns with the intrinsic statistics of the data (Ilyas et al., 2019; Allen-Zhu & Li, 2022). Consequently, these perturbations behave more like genuine features than noise, making them challenging to be modeled explicitly and prone to being inadvertently reconstructed. To address this issue, we first explore the distribution changes of perturbations during the optimization process and initially mitigate their impact through progressive downsampling. Building upon these insights, we propose a coarse-to-fine TN incremental learning algorithm and introduce a novel adversarial optimization objective to avoid overly constraining the reconstruction error, preventing inadvertently restoring adversarial perturbations. Unlike classical TN methods applied to adversarial examples, our coarse-to-fine TN method prevents naive low-rank representation of the input and encourages the reconstructed examples to approximate the unobserved clean examples.

We empirically evaluate the performance of TNP by comparing it with AT and AP across attack settings using multiple classifiers on CIFAR-10, CIFAR-100, and ImageNet. The results demonstrate that TNP achieves robustness with strong generalization across diverse adversarial scenarios. Specifically, TNP achieved a 26.45% improvement in average robust accuracy over AT across different norm threats, a 9.39% improvement over AP across multiple attacks, and a 6.47% improvement over AP across different datasets. Furthermore, in denoising tasks, TNP effectively removes adversarial perturbations while preserving consistency between the reconstructed clean example and the reconstructed adversarial example. These results collectively underscore the effectiveness and potential of TNP. In summary, our contributions are as follows:

- We propose a model-free optimization-based technique based on tensor network representation, which requires neither a powerful generative model nor reliance on specific dataset distributions, making it a general-purpose adversarial purification.

- Based on our analysis of the distribution changes of adversarial perturbations during optimization, we design a novel adversarial optimization objective for coarse-to-fine TN representation learning to prevent the restoration of adversarial perturbations.

- We conduct extensive experiments on various datasets, demonstrating that our method achieves state-of-the-art performance, especially exhibiting strong generalization across diverse adversarial scenarios.

## 2 Related Works

**Adversarial robustness**  To defend against adversarial attacks, researchers have developed various techniques aimed at enhancing the robustness of DNNs. Goodfellow et al. (2015) propose AT technique to defend against adversarial attacks by retraining classifiers with adversarial examples (Wang et al., 2019; Tack et al., 2022). In contrast, AP methods (Shi et al., 2021; Srinivasan et al., 2021) aim to purify adversarial examples before classification without retraining the classifier. Currently, the most common AP methods (Nie et al., 2022; Bai et al., 2024) rely on pre-trained generative models as purifiers, which are trained on specific datasets and hard to generalize to data distributions outside their training domain. Lin et al. (2024a) propose applying AT (Zhang et al., 2019) technique to AP,

optimizing the purifier to adapt to new data distributions, at the cost of substantial training costs. Although TNP employs AP technique, it fundamentally differs from these works in that a model-free optimization-based framework relying solely on the information of the single input example for AP, without requiring any additional priors from pre-trained models and training costs.

**Tensor network and TN-based defense methods** Tensor network (TN) is a classical tool in signal processing, with many successful applications in image completion and denoising (Kolda & Bader, 2009; Cichocki et al., 2015). Compared to classical TN methods such as TT (Oseledets, 2011) and TR (Zhao et al., 2016), we employ the quantized technique (Khoromskij, 2011) and develop a coarse-to-fine strategy. Recent work (PuTT, Loeschcke et al., 2024) also employs a coarse-to-fine strategy, aiming to achieve better initialization for faster and more efficient TT decomposition by minimizing the reconstruction error. In comparison, our method progresses from low to high resolution, explicitly targeting perturbation removal and analyzing the impact of downsampling on perturbations. Furthermore, we propose a novel optimization objective that goes beyond simply minimizing the reconstruction error, focusing instead on preventing the restoration of perturbations.

With the growing concern over adversarial robustness, a line of work has attempted to leverage TNs as robust denoisers to defend against adversarial attacks. In particular, Yang et al. (2019) reconstruct images and retrain classifiers to adapt to the new reconstructed distribution. Entezari & Papalexakis (2022) analyze vanilla TNs and show their effectiveness in removing high-frequency perturbations. Additionally, (Bhattarai et al., 2023) extend the application of TNs beyond data to include classifiers, a concept similar to the approaches of (Rudkiewicz et al., 2024; Phan et al., 2023). Furthermore, (Song et al., 2024) employ training-free techniques while incorporating ground truth information to defend against adversarial attacks. However, the aforementioned methods rely on additional prior or are limited to specific attacks. In this paper, we aim to achieve robustness solely by optimizing TNs themselves, establishing them as a plug-and-play and promising adversarial purification technique.

## 3 Backgrounds

**Notations** Throughout the paper, we denote scalars, vectors, matrices, and tensors as lowercase letters, bold lowercase letters, bold capital letters, and calligraphic bold capital letters, e.g., $x$, $\boldsymbol{x}$, $\boldsymbol{X}$ and $\boldsymbol{\mathcal{X}}$, respectively. A $D$-order tensor is an $D$-dimensional array, e.g., a vector is a 1st-order tensor and a matrix is a 2nd-order tensor. For a $D$-order tensor $\boldsymbol{\mathcal{X}} \in \mathbb{R}^{I_1 \times \cdots \times I_D}$, we denote its $(i_1, \ldots, i_D)$-th entry as $x_{\mathbf{i}}$, where $\mathbf{i} = (i_1, \ldots, i_D)$. Following the conventions in deep learning, we treat images as vectors, e.g., input example $\boldsymbol{x}_{in}$, clean example $\boldsymbol{x}_{cln}$, adversarial example $\boldsymbol{x}_{adv}$ and reconstructed example $\boldsymbol{y}$.

**Tensor network decomposition** Given a $D$-order tensor $\boldsymbol{\mathcal{X}} \in \mathbb{R}^{I_1 \times \cdots \times I_D}$, tensor network decomposition factorizes $\boldsymbol{\mathcal{X}}$ into $D$ smaller latent components by using some predefined tensor contraction rules. Among tensor network decompositions, Tensor Train (TT) decomposition (Oseledets, 2011) enjoys both quasi-optimal approximation as well as the high compression rate of large and complex data tensors. In particular, a $D$-order tensor $\boldsymbol{\mathcal{X}} \in \mathbb{R}^{I_1 \times \cdots \times I_D}$ has the TT format as $x_{\mathbf{i}} = \boldsymbol{A}_{i_1}^1 \boldsymbol{A}_{i_2}^2 \ldots \boldsymbol{A}_{i_D}^D$, where $\boldsymbol{A}_{i_d}^d \in \mathbb{R}^{r_{d-1} \times r_d}$, for $d \in [D]$ and $i_d \in [I_d]$. Then, $(1, r_1, \ldots, r_{d-1}, 1)$ is the TT rank of $\boldsymbol{\mathcal{X}}$. For simplicity, we denote $\boldsymbol{\mathcal{X}} = \text{TT}(\boldsymbol{\mathcal{A}}^1, \ldots, \boldsymbol{\mathcal{A}}^D)$. When each dimension $I_d$ of $\boldsymbol{\mathcal{X}}$ is large, quantized tensor train (QTT, Khoromskij, 2011) becomes highly efficient, which splits each dimension in powers of two. For example, a $2^D \times 2^D$ image can be rearranged into a more expressive and balanced $D$-order tensor. For brevity, hereafter, a $2^D \times 2^D$ image $\boldsymbol{x}_D$ shall be called a resolution $D$ image, whose quantized tensor is $\boldsymbol{\mathcal{X}}_D = \text{Q}(\boldsymbol{x}_D)$. QTT core denotes the core tensor after decomposition.

## 4 Method

Tensor network (TN) is a classical tool in signal processing, with many successful applications in image completion and denoising. By leveraging the $\ell_2$-norm as the primary optimization criterion, which aligns well with the statistical properties of a normal distribution, these methods (Phan et al., 2020; Loeschcke et al., 2024) have demonstrated strong capabilities in removing Gaussian noise.

However, the distribution of well-designed adversarial perturbations is essentially different from Gaussian noise and cannot be modeled explicitly (Ilyas et al., 2019; Allen-Zhu & Li, 2022), which challenges the conventional assumptions of TN-based denoising methods, leading to ineffectiveness

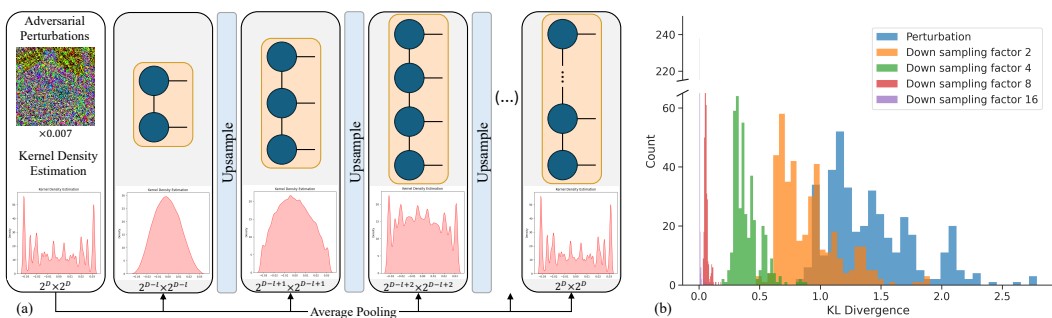

Figure 1: Compare the adversarial perturbations in the downsampled images. (a) The distribution changes of adversarial perturbations during downsampling process. (b) The KL divergence between the adversarial perturbations and the Gaussian distributions with the same sample mean and variance.

on adversarial purification for $\boldsymbol{x}_{adv}$. To minimize the loss $\|\boldsymbol{x}_{adv} - \text{TN}(\boldsymbol{x}_{adv})\|_2$, TN decompositions fit all feature components of $\boldsymbol{x}_{adv}$, including the adversarial perturbations. However, in the presence of adversarial attacks, we aim to restore unobserved $\boldsymbol{x}_{cln}$ from the input $\boldsymbol{x}_{adv}$, that is: $\text{TN}(\boldsymbol{x}_{adv}) \approx \boldsymbol{x}_{cln}$ rather than $\boldsymbol{x}_{adv}$. Based on the above analysis, it is crucial to overcome two challenges in designing an effective TN method: *Q1. How can we transform the non-specific adversarial perturbations into a form amenable to TN modeling? Q2. How can we avoid overly constraining the reconstruction error from inadvertently restoring those perturbations?*

For *Q1*, we explore how adversarial perturbations behave under downsampling with average pooling. Intuitively, the central limit theorem suggests that as an image is progressively downsampled, aggregated perturbations begin to resemble a normal distribution. Thus, even an $\ell_2$-based penalty becomes effective in suppressing the perturbations at coarse resolution.

However, while this insight helps suppress perturbations at lower resolutions, there remains the challenge of reconstructing the original resolution image. When upsampling and further optimizing using $\|\boldsymbol{x}_{adv} - \text{TN}(\boldsymbol{x}_{adv})\|_2$, the perturbations will still be restored. This connects with *Q2*, for which we design a new optimization objective.

## 4.1 Downsampling using average pooling

An intuitive explanation for why downsampling aids in perturbation removal can be derived from the Central Limit Theorem (CLT, Grzenda & Zieba, 2008). When an image is downsampled by average pooling, the random components (e.g., pixel-level noise or minor adversarial perturbations) within those pooling patches are aggregated. We hypothesize that, given an adversarial example $\boldsymbol{x}_{adv}$, downsampling the $\boldsymbol{x}_{adv}$ from its original resolution $D$ to a lower resolution $D - 1$ will smooth out the perturbations. As the downsampling process progresses further, the distribution of the aggregated perturbations in the coarse resolution image $\boldsymbol{x}_{D-l}$ is expected to converge toward a normal distribution, as illustrated in Figure 1a. More results are shown in Appendix G.

To investigate this hypothesis in real datasets, we measure the KL divergence between the histograms of adversarial perturbations and the Gaussian distributions with the same sample mean and variance across 512 images from ImageNet. As shown in Figure 1b, the distribution of those perturbations progressively aligns with that of Gaussian noise as the downsampling process progresses. Consequently, even classical TN methods can effectively remove or mitigate adversarial perturbations at coarse resolution. Additionally, we further compare the influence of different downsampling methods to underscore the advantages of average pooling, as discussed in Appendix A.

## 4.2 Tensor network purification

Building upon our downsampling-based intuition, we design a coarse-to-fine purification pipeline by extending PuTT (Loeschcke et al., 2024), which employs progressive downsampling for better initialization of QTT cores. The workflow of tensor network purification (TNP) for classification tasks is illustrated in Figure 2, where the quantized $\boldsymbol{\mathcal{X}} = \text{Q}(\boldsymbol{x})$, TT decomposition $\boldsymbol{\mathcal{X}} \approx \boldsymbol{\mathcal{Y}} = \text{TT}(\boldsymbol{\mathcal{A}}^1, \ldots, \boldsymbol{\mathcal{A}}^D)$, and reconstruction $\boldsymbol{y} = \text{Q}^{-1}(\boldsymbol{\mathcal{Y}})$ processes are depicted.

**Algorithm 1** Adversarial optimization process.

**Input:** Example $\boldsymbol{x}_d$, number of iterations $T$, steps $N$, scale $\alpha$ and $\eta$, learning rate $\beta$
Initialize $\boldsymbol{y}_d \leftarrow \mathrm{P}_d(\boldsymbol{y}_{d-1}), \boldsymbol{\delta}_d \leftarrow \boldsymbol{0}$
**for** $t = 1, 2, \ldots, T$ **do**
    **for** $n = 1, 2, \ldots, N$ **do**
        $\ell \leftarrow \mathcal{L}_{adv}(\boldsymbol{y}_d + \boldsymbol{\delta}_d, \boldsymbol{x}_d)$
        $\boldsymbol{\delta}_d \leftarrow \mathrm{clip}(\boldsymbol{\delta}_d + \alpha\,\mathrm{sign}(\nabla_{\boldsymbol{y}_d}\ell), -\eta, \eta)$
    $\boldsymbol{\delta}_d^* \leftarrow \mathrm{clip}(\boldsymbol{y}_d + \boldsymbol{\delta}_d, 0, 1) - \boldsymbol{y}_d$
    Gradient descent based on Eq. (1):
    $\boldsymbol{y}_d \leftarrow \boldsymbol{y}_d - \beta\nabla_{\boldsymbol{y}_d}\mathcal{L}_{tnp}(\boldsymbol{x}_d, \boldsymbol{y}_d, \boldsymbol{\delta}_d^*)$
**return** $\boldsymbol{y}_d$

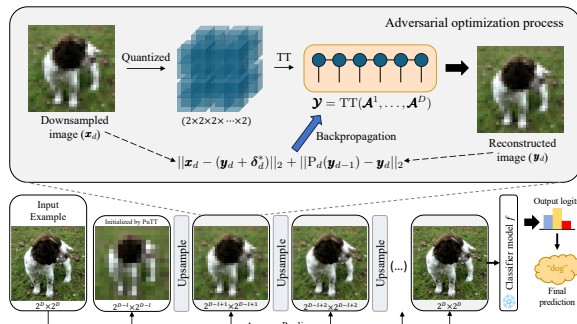

Figure 2: Illustration of tensor network purification.

Initially, the $2^D \times 2^D$ input example $\boldsymbol{x}_D$ (potentially adversarial example $\boldsymbol{x}_{adv}$ or clean example $\boldsymbol{x}_{cln}$), whose quantized version is a $D$-order tensor $\boldsymbol{\mathcal{X}}_D$, is first downsampled to a resolution $D - l$ example $\boldsymbol{x}_{D-l}$, corresponding to a $(D - l)$-order tensor $\boldsymbol{\mathcal{X}}_{D-l}$. The QTT cores of $\boldsymbol{\mathcal{X}}_{D-l}$ are optimized by PuTT via backpropagation within a standard reconstruction error $||\boldsymbol{x}_{D-l} - \boldsymbol{y}_{D-l}||_2$. Once the approximation of $\boldsymbol{\mathcal{X}}_{D-l}$ is stabilized, the prolongation operator $\boldsymbol{\mathcal{P}}_{D-l+1}$ is applied to the QTT format of $\boldsymbol{\mathcal{X}}_{D-l}$, producing a $(D - l + 1)$-order tensor $\boldsymbol{\mathcal{P}}_{D-l+1}\boldsymbol{\mathcal{X}}_{D-l}$. Additionally, we define the linear function $\mathrm{P}_d(\cdot)$ acts on the image level, with the effect of upsampling from resolution $d - 1$ to $d$, details in Appendix B.2. This serves as an initialization to find the optimal QTT cores of $\boldsymbol{\mathcal{X}}_{D-l+1}$ and reconstructed downsampled example $\boldsymbol{y}_{D-l}$.

Next, the input example $\boldsymbol{x}_D$ is once again downsampled to a resolution $D - l + 1$ example $\boldsymbol{x}_{D-l+1}$. At this stage, the QTT cores of $\boldsymbol{\mathcal{X}}_{D-l+1}$ are optimized using the adversarial optimization objective within a novel loss function as shown in Eq. (1). Similarly, once the approximation of $\boldsymbol{\mathcal{X}}_{D-l+1}$ stabilizes, the upsampling operation is performed. This process is repeated iteratively until reaching the QTT approximation $\boldsymbol{\mathcal{Y}}_D$ of the original resolution $\boldsymbol{\mathcal{X}}_D$.

Finally, TNP can purify potential adversarial examples ($\boldsymbol{x}_{cln}$ or $\boldsymbol{x}_{adv}$) before feeding them into classifier $f$, e.g., $f(\mathrm{TNP}(\boldsymbol{x}_{cln})) = f(\mathrm{TNP}(\boldsymbol{x}_{adv})) = gt$, where $gt$ is the ground truth label. As a plug-and-play module, TNP requires no modification to $f$ and can be integrated with any classifier.

### 4.3 Adversarial optimization process

Following the coarse-to-fine process, despite the downsampling with average pooling and subsequent PuTT at lower resolutions can mitigate adversarial perturbations, the other challenge arises upon reconstructing the image at the original resolution, where minimizing the standard reconstruction error will inevitably restore the adversarial perturbations.

Unlike traditional reconstruction, in the context of adversarial attacks, we can only observe the adversarial example $\boldsymbol{x}_{adv}$, while the goal is to reconstruct a "clean" $\boldsymbol{y}$ closing to the unobserved clean example $\boldsymbol{x}_{cln}$. To bridge the gap between $\boldsymbol{x}_{adv}$ and $\boldsymbol{x}_{cln}$, we propose a new optimization objective that introduces an auxiliary variable $\boldsymbol{\delta}$. Moreover, we leverage the previously reconstructed downsampled example as a crucial prior to guide the approximation toward $\boldsymbol{x}_{cln}$.

Here, we outline the optimization procedure for $\boldsymbol{x}_d$, which corresponds to the gray box in Figure 2. Formally, given the resolution $d$ example $\boldsymbol{x}_d$, we attempt to obtain the reconstructed example $\boldsymbol{y}_d$ by performing gradient descent on optimization loss functions of

$$\mathcal{L}_{tnp}(\boldsymbol{x}_d, \boldsymbol{y}_d, \boldsymbol{\delta}_d^*) = ||\boldsymbol{x}_d - (\boldsymbol{y}_d + \boldsymbol{\delta}_d^*)||_2 + ||\mathrm{P}_d(\boldsymbol{y}_{d-1}) - \boldsymbol{y}_d||_2,$$
$$\text{s.t. } \boldsymbol{\delta}_d^* = \arg\max_{||\boldsymbol{\delta}_d|| < \eta} \mathcal{L}_{adv}(\boldsymbol{y}_d + \boldsymbol{\delta}_d, \boldsymbol{x}_d), \tag{1}$$

where $d \in [D - l + 1, D]$ and $\eta$ is a scale hyperparameter.

The auxiliary variable $\boldsymbol{\delta}^*$ is determined through an inner maximization process that utilizes a non-convex loss function $\mathcal{L}_{adv}$. We employ a perceptual metric, structural similarity index measure (SSIM, Hore & Ziou, 2010), as $\mathcal{L}_{adv}$ to explore more potential solutions and better handle complex perturbation patterns. While $\boldsymbol{\delta}^*$ does not exactly represent the true adversarial perturbation, bounding

$\|\boldsymbol{\delta}\| < \eta$ can partially ensure that the misalignment between $\boldsymbol{y}$ and $\boldsymbol{x}_{adv}$ remains controlled, effectively ensuring that $\boldsymbol{y}$ does not simply collapse into the adversarial example $\boldsymbol{x}_{adv}$.

However, precisely because $\boldsymbol{\delta}^*$ does not represent the true perturbation, minimizing $\|\boldsymbol{x}_d - (\boldsymbol{y}_d + \boldsymbol{\delta}_d^*)\|_2$ may not yield the desired clean example. To address this limitation, we introduce a second loss term $\|\mathrm{P}_d(\boldsymbol{y}_{d-1}) - \boldsymbol{y}_d\|_2$, which serves as a crucial "prior". Specifically, we utilize the reconstructed downsampled example $\boldsymbol{y}_{d-1}$ as an additional constraint to aid in approximating the $\boldsymbol{x}_{cln}$. Building upon the observations in Figure 1, we start from the resolution $D - l$ example $\boldsymbol{x}_{D-l}$ that is optimized by PuTT, and then perform upsampling to the higher resolution to produce a clean-leaning reference, which acts to nudge $\boldsymbol{y}$ toward a less perturbed distribution. Although we never have direct access to the true clean example $\boldsymbol{x}_{cln}$, our loss provides an effective surrogate prior and guides the optimization process. The detailed algorithm of our adversarial optimization process is shown in Algorithm 1.

## 5    Experiments

In this section, we conduct comprehensive experiments on multiple datasets across various settings. The classification results demonstrate that TNP achieves robustness with strong generalization. We further investigate the removal of adversarial perturbations using tensor network decompositions and find that only TNP effectively removes the perturbations while preserving consistency between clean and adversarial examples. These results collectively highlight the effectiveness and potential of TNP.

### 5.1    Experimental setup

**Datasets and model architectures**   We conduct extensive experiments on CIFAR-10, CIFAR-100 (Krizhevsky et al., 2009) and ImageNet (Deng et al., 2009) to empirically validate the effectiveness of the proposed methods against adversarial attacks. For classification tasks, we utilize the pre-trained ResNet (He et al., 2016) and WideResNet (Zagoruyko & Komodakis, 2016) models.

**Adversarial attacks**   We evaluate our method against AutoAttack (Croce & Hein, 2020), a widely used benchmark that integrates both white-box and black-box attacks. Additionally, following the guidance of Lee & Kim (2023), we utilize PGD (Madry et al., 2018) with EOT (Athalye et al., 2018b) for a more comprehensive evaluation. Considering the potential robustness overestimation caused by obfuscated gradients of the purifier model, we utilize BPDA (Athalye et al., 2018a) as an adaptive attack with the knowledge of both purifier and classifier, following the setting by Yang et al. (2019); Lin et al. (2024a). Further implementation details and discussion are provided in Appendix C.

**Compared methods**   We conduct experiments on the common benchmark and compare the robustness of our method with those listed in RobustBench (Croce et al., 2021). We evaluate the generalization of existing defense methods, including AT methods (Gowal et al., 2020, 2021; Laidlaw et al., 2021; Dolatabadi et al., 2022; Pang et al., 2022) and AP methods, with particular attention to diffusion-based AP (Yoon et al., 2021; Nie et al., 2022; Lee & Kim, 2023; Lin et al., 2024b). Furthermore, we include comparisons with Tensor Train (TT, Oseledets, 2011), Tensor Ring (TR, Zhao et al., 2016), quantized technique (Khoromskij, 2011) and PuTT (Loeschcke et al., 2024).

Due to the high computational cost of evaluating methods with multiple attacks, following the guidance of Nie et al. (2022), we randomly select 512 images from the test set for robust evaluation. All experiments presented in the paper are conducted by NVIDIA RTX A5000 with 24GB GPU memory, CUDA v11.7, and cuDNN v8.5.0 in PyTorch v1.13.11. More details in Appendix D.

### 5.2    Robustness comparison on RobustBench

In this section, we evaluate our method for defending against AutoAttack and compare it with the methods under all adversarial settings listed in RobustBench (Croce et al., 2021). Tables 1 to 4 present the performance of various defense methods against $l_\infty$ ($\epsilon = 8/255$) and $l_2$ ($\epsilon = 0.5$) threats. Overall, the highest robust accuracy achievable by our method is generally on par with existing methods without using extra data (the dataset introduced by Carmon et al. (2019)). Specifically, compared to the second-best method, our method improves the robust accuracy by 1.67% on CIFAR-100, by 1.84% on ImageNet, and the average robust accuracy by 0.36% on CIFAR-10.

Due to the overfitting of WideResNet-28-10 trained on the limited data available in CIFAR-10, we observe that the results with standard classifier (Ours) struggle to reach state-of-the-art performance,

Table 1: Standard and robust accuracy against AutoAttack $l_\infty$ threat ($\epsilon = 8/255$) on CIFAR-10. ([†]the methods use additional synthetic images.)

| Defense method | Extra data | Standard Acc. | Robust Acc. |
|---|---|---|---|
| Gowal et al. (2020) | ✓ | 89.48 | 62.70 |
| Bai et al. (2023) | ✓[†] | 95.23 | 68.06 |
| Chen & Lee (2024) | × | 86.10 | 58.09 |
| Cui et al. (2024) | ×[†] | 92.16 | 67.73 |
| Nie et al. (2022) | × | 89.02 | 70.64 |
| Zhang et al. (2024) | × | 90.04 | 73.05 |
| Lin et al. (2024a) | × | 90.62 | 72.85 |
| Ours | × | 82.23 | 55.27 |
| Ours* | × | 91.99 | 72.85 |

Table 2: Standard and robust accuracy against AutoAttack $l_2$ threat ($\epsilon = 0.5$) on CIFAR-10.

| Defense method | Extra data | Standard Acc. | Robust Acc. |
|---|---|---|---|
| Augustin et al. (2020) | ✓ | 92.23 | 77.93 |
| Gowal et al. (2020) | ✓ | 94.74 | 80.53 |
| Wang et al. (2023) | ×[†] | 95.16 | 83.68 |
| Rebuffi et al. (2021) | ×[†] | 91.79 | 78.32 |
| Ding et al. (2019) | × | 88.02 | 67.77 |
| Nie et al. (2022) | × | 91.03 | 78.58 |
| Ours | × | 82.23 | 68.16 |
| Ours* | × | 91.99 | 79.49 |

Table 3: Standard and robust accuracy against AutoAttack $l_\infty$ ($\epsilon = 8/255$) on CIFAR-100.

| Defense method | Extra data | Standard Acc. | Robust Acc. |
|---|---|---|---|
| Hendrycks et al. (2019) | ✓ | 59.23 | 28.42 |
| Debenedetti et al. (2023) | ✓ | 70.76 | 35.08 |
| Cui et al. (2024) | ×[†] | 73.85 | 39.18 |
| Wang et al. (2023) | ×[†] | 75.22 | 42.67 |
| Pang et al. (2022) | × | 63.66 | 31.08 |
| Jia et al. (2022) | × | 67.31 | 31.91 |
| Ours | × | 62.30 | 44.34 |

Table 4: Standard and robust accuracy against AutoAttack $l_\infty$ threat ($\epsilon = 4/255$) on ImageNet.

| Defense method | Extra data | Standard Acc. | Robust Acc. |
|---|---|---|---|
| Salman et al. (2020) | × | 64.02 | 37.89 |
| Bai et al. (2021) | × | 67.38 | 35.51 |
| Nie et al. (2022) | × | 67.79 | 40.93 |
| Bai et al. (2024) | × | 70.41 | 41.70 |
| Chen & Lee (2024) | × | 68.76 | 40.60 |
| Ours | × | 65.43 | 42.77 |

consistent with findings from Chen & Lee (2024). To further improve robust accuracy, most AT methods incorporate additional synthetic data to train a robust classifier. Following this, we conduct experiments with the robust classifier (Ours*), which utilizes an additional 20M synthetic images in training (Cui et al., 2024). This leads to a significant improvement in robust accuracy on CIFAR-10. Moreover, compared to the used robust classifier (Cui et al., 2024), our method further improves the robust accuracy by 5.12%. These results are consistent across multiple datasets and norm threats, confirming the effectiveness of our method and its potential for defending against adversarial attacks.

## 5.3 Generalization comparison across various adversarial scenarios

As previously highlighted, the existing defense methods are often criticized for their lack of generalization across different norm threats, attacks, and datasets. In the following, we evaluate the performance of our method under various adversarial settings to demonstrate its robust generalization.

Table 5: Standard accuracy and robust accuracy against AutoAttack $l_\infty$ ($\epsilon = 8/255$) and $l_2$ ($\epsilon = 1.0$) threats on CIFAR-10 with ResNet-50.

| Type | Defense method | SA | Robust Acc. | |
|---|---|---|---|---|
| | | | AA $l_\infty$ | AA $l_2$ |
| | Standard Training | 94.8 | 0.0 | 0.0 |
| AT | Training with $l_\infty$ | 86.8 | 49.0 | 19.2 |
| | Training with $l_2$ | 85.0 | 39.5 | 47.8 |
| | Laidlaw et al. (2021) | 82.4 | 30.2 | 34.9 |
| | Dolatabadi et al. (2022) | 83.2 | 40.0 | 33.9 |
| AP | Nie et al. (2022) | 88.2 | 70.0 | 70.9 |
| | Lin et al. (2024a) | 89.1 | 71.2 | 73.4 |
| | Ours | 88.3 | 73.2 | 67.0 |

**Results analysis on different norm threats** Table 5 shows that AT methods (Laidlaw et al., 2021; Dolatabadi et al., 2022) are limited in defending against unseen attacks and can only effectively be against the specific attacks they are trained on. An intuitive idea is to apply AT across all norm threats or develop more general constraints to obtain a robust model. However, training such a model is challenging due to the inherent differences among various attacks. In contrast, AP methods (Nie et al., 2022; Lin et al., 2024a) exhibit strong generalization, effectively defending against unseen attacks. The results demonstrate that our method also possesses strong generalization capabilities against unseen attacks, achieving performance close to

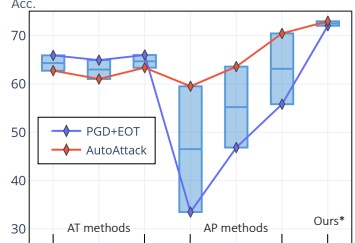

Figure 3: Comparison of robust accuracy against multiple attacks.

Table 6: Standard accuracy (SA) and robust accuracy (RA) against AutoAttack $l_\infty$ ($\epsilon = 8/255$) on CIFAR-10 and CIFAR-100. The pre-trained generative model used in AP is trained on CIFAR-10.

| Method | CIFAR-10 | | CIFAR-100 | | Avg. | |
|---|---|---|---|---|---|---|
| | SA | RA | SA | RA | SA | RA |
| Standard | 94.78 | 0.00 | 81.86 | 0.00 | 88.32 | 0.00 |
| AT | 92.16 | 67.73 | 73.85 | 39.18 | 83.01 | 53.46 |
| AP | 89.02 | 70.64 | 38.09 | 33.79 | 63.56 | 52.22 |
| Ours* | 91.99 | 72.85 | 71.48 | 44.53 | 81.74 | 58.69 |

the AP methods while significantly outperforming the existing AT methods. Specifically, compared to the best AT method, our method improves average robust accuracy by 26.45%.

**Results analysis on multiple attacks** Figure 3 shows the comparison of robust accuracy against PGD+EOT and AutoAttack with $l_\infty$ ($\epsilon = 8/255$) threat on CIFAR-10 with WideResNet-28-10. When facing different attacks within the same threat, AT methods (Gowal et al., 2020, 2021; Pang et al., 2022) exhibit better generalization than AP methods (Yoon et al., 2021; Nie et al., 2022; Lee & Kim, 2023). Typically, robustness evaluation is based on the worst-case results of the robust accuracy. Under this criterion, our method outperforms all AT and AP methods. Specifically, compared to the best AP method, our method improves average robust accuracy by 9.39%.

**Results analysis on different datasets** Table 6 shows the generalization of the methods across different datasets. As previously highlighted, the existing AP methods typically rely on specific datasets. For AP method, when a pre-trained generative model trained on CIFAR-10 is applied to adversarial robustness evaluation on CIFAR-100, both standard accuracy and robust accuracy drop significantly. This occurs because the pre-trained generative model can only generate the data it has learned. Although the input examples originate from CIFAR-100, the generative model attempts to output one of the ten classes from CIFAR-10, severely distorting the semantic information of the input examples and leading to low classification accuracy. In contrast, our method exhibits strong generalization across different datasets, achieving comparable robust performance on CIFAR-100 as on CIFAR-10. Specifically, compared to the AP method (Nie et al., 2022), our method improves the average robust accuracy by 6.47%.

Unlike existing methods, TNP employs an optimization-based strategy that operates solely on the given input, without relying on prior knowledge learned from large-scale training datasets or strong assumptions about attacks, thereby retaining strong generalization across various scenarios.

## 5.4 Denoising tasks

In this section, we evaluate the effectiveness of our method on non-classification tasks through visual comparisons and various quantitative metrics.

**Ablation study** Figure 4 shows the comparison of visualizations on ImageNet. The top row in (a) displays the input clean example (CE), and its corresponding reconstructed clean examples (rec. CE) generated by traditional $\ell_2$ loss $||\boldsymbol{x} - \boldsymbol{y}||_2$ and our proposed loss function, while (b) displays the

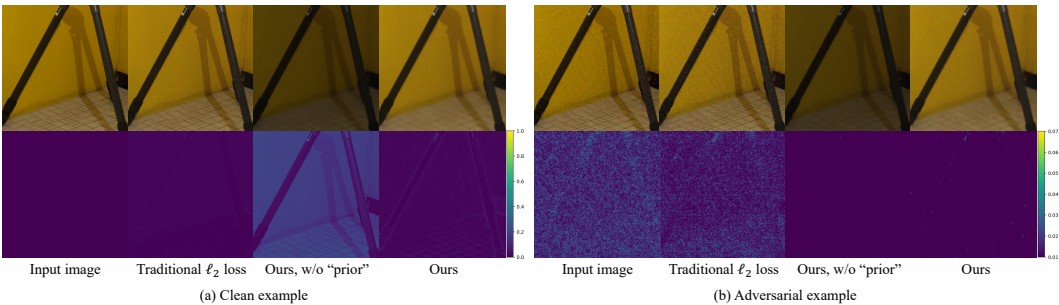

Figure 4: Comparison of visualizations. The original input image and corresponding reconstructed image (top), along with the error maps (bottom) for the clean example and the adversarial example.

Table 7: Comparisons on CIFAR-10. The rec. CEs are expected to closely match the CEs, whereas the rec. AEs should remain sufficiently different from the AEs to avoid restoring perturbations.

| Defense method | CLN: CEs & rec.CEs | | | | ADV: AEs & rec.AEs | | | | REC: rec.CEs & rec.AEs | | |
|---|---|---|---|---|---|---|---|---|---|---|---|
| | Acc. | NRMSE | SSIM | PSNR | Acc. | NRMSE | SSIM | PSNR | NRMSE | SSIM | PSNR |
| Standard | 94.78 | - | - | - | 0.00 | - | - | - | - | - | - |
| TT | 87.30 | 0.0507 | 0.9526 | 31.14 | 36.13 | 0.0650 | 0.8977 | 28.99 | 0.0267 | 0.9790 | 39.10 |
| TR | **94.34** | **0.0171** | **0.9938** | **40.58** | 0.98 | 0.0464 | 0.9210 | 31.91 | 0.0322 | 0.9598 | 35.51 |
| QTT | 84.57 | 0.0613 | 0.9253 | 29.49 | 51.56 | 0.0724 | 0.8808 | 28.06 | 0.0233 | 0.9855 | 39.88 |
| QTR | 83.40 | 0.0613 | 0.9254 | 29.49 | 49.41 | 0.0724 | 0.8785 | 28.06 | 0.0231 | 0.9853 | 39.96 |
| PuTT | 80.86 | 0.0626 | 0.9261 | 29.32 | 44.14 | 0.0742 | 0.8787 | 27.84 | 0.0311 | 0.9770 | 38.03 |
| Ours | 82.23 | 0.0644 | 0.9203 | 29.06 | **55.27** | **0.0748** | **0.8707** | **27.77** | **0.0218** | **0.9863** | **40.37** |

reconstructed adversarial examples (rec. AE) for the input adversarial example (AE). Additionally, we create error maps to highlight differences, which (a) between the rec. CEs and the input CEs, and (b) between the rec. AEs and the rec. CEs, as shown at the bottom of Figure 4. The results indicate that while our method does not match the classical TN methods in reconstructing CEs, it significantly outperforms them in removing adversarial perturbations from AEs.

Specifically, when processing CEs, the rec. examples generated by traditional $\ell_2$ loss are almost identical to the original ones, whereas our method is slightly less effective in restoring some details. However, when processing AEs, the rec. examples from traditional $\ell_2$ loss remain consistent with the original ones, leading to the preservation of adversarial perturbations, as highlighted in Figure 4b. In contrast, our method better removes those perturbations, ensuring that the rec. AEs and the rec. CEs retain similar information. Moreover, we evaluate the necessity of the second term in Eq. (1), which serves as a surrogate prior constraint to optimize the reconstructed examples toward the clean data distribution. As observed, removing this constraint eliminates prior information from the optimization process, increasing the likelihood of significant deviation in the wrong direction.

**Quantitative results analysis** Table 7 shows the quantitative results of the denoising task for AEs and CEs, with detailed descriptions of evaluation metrics provided in Appendix D.2. We compare our method with existing tensor network decompositions, including TT, TR, QTT, QTR, and PuTT. While our method does not achieve the best denoising performance on clean examples, it still maintains classification performance well, achieving 82.23% standard accuracy with vanilla WideResNet-28-10. More importantly, our method outperforms others in the next two columns. Specifically, when processing AEs, our method yields the highest NRMSE and the lowest SSIM and PSNR, achieving the highest robust accuracy. This outcome is expected, as our goal is to ensure that the rec. AEs differ from the original AEs (i.e., lower SSIM and PSNR, and higher NRMSE in the "ADV" column) while rec. AEs closely resembling the rec. CEs (i.e., higher SSIM and PSNR, and lower NRMSE in the "REC" column). These results align well with the visual observations in Figure 4 and consistently demonstrate the effectiveness of our method, highlighting its potential in adversarial scenarios.

**Limitations and future works** We identify several open problems related to TNP: (1) Although TNP is a training-free technique, it incurs additional optimization costs during inference, which poses challenges for deployment in low-latency scenarios, see more discussion in Appendices E.1 and E.2. (2) As a model-free optimization-based technique, TNP is inherently more resistant to adaptive attacks, see more discussion in Appendix C. Accordingly, developing more advanced optimization strategies and adaptive attack strategies specifically tailored to TNP remains a valuable direction for future research. We hope that our work will motivate further exploration of these challenges.

# 6 Conclusion

In this paper, we propose a novel model-free optimization-based adversarial purification (AP) built upon a specially designed tensor network decomposition. Extensive experiments on CIFAR-10, CIFAR-100, and ImageNet demonstrate that our method (TNP) achieves state-of-the-art performance with strong generalization across diverse scenarios. Additionally, we further identify several open challenges related to TNP, and believe that continued exploration of TN–based purification remains an exciting research direction for developing a plug-and-play and effective AP technique.

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

 **Appendix**

 ## A   Influence of different sampling methods

To support our hypothesis of using the average pooling, we test it with stride sampling, which selects pixels with constant strides. In principle, the stride sampling would not change the distribution of perturbations. Therefore, it serves as a baseline to compare the influence of distributions.

We test four types of noise distributions: (1) Gaussian $\mathcal{N}(0, 0.3^2)$, (2) Mixture of Gaussian (MoG), $0.5 \cdot \mathcal{N}(-1.0, 0.5^2) + 0.5 \cdot \mathcal{N}(1.0, 0.5^2)$, (3) Beta distribution, $\text{Beta}(0.5, 0.5) - 0.5$, and (4) Uniform distribution, $\text{Uniform}(-0.5, 0.5)$. For MoG, Beta and uniform noises, we scale them to have the same signal-to-noise ratio with the Gaussian distribution. We add the noises on the Girl image (Loeschcke et al., 2024) with resolution $1024 \times 1024$. First, we show the noise distributions in Figure 5. As can be seen, the Avg Pooling strategy transforms the non-Gaussian noises into Gaussian-like noises, while the Stride sampling would not. Second, we run the PuTT algorithm with different sampling methods for 100 times. The violin plot of denoising results are shown in Figure 6. In Gaussian distribution, the Stride sampling is better than AvgPooling. While for non-Gaussian noises, the AvgPooling is more robust and better than Stride. The denoising results indicate that the average pooling can handle different types of noises, which is consistent with our hypothesis. However, as we introduced, this might not be enough, since we need to deal with the original image and noises in the final stage.

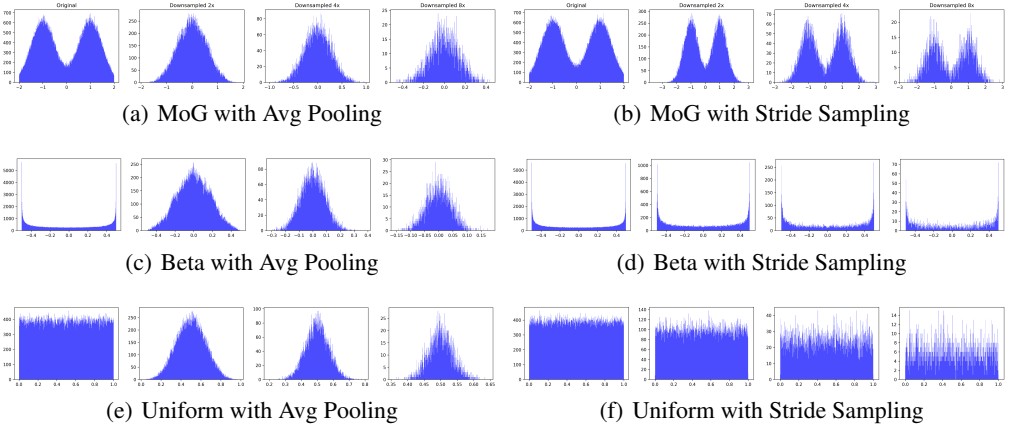

(a) MoG with Avg Pooling  (b) MoG with Stride Sampling

(c) Beta with Avg Pooling  (d) Beta with Stride Sampling

(e) Uniform with Avg Pooling  (f) Uniform with Stride Sampling

Figure 5: Histogram figures of noises under different sampling methods.

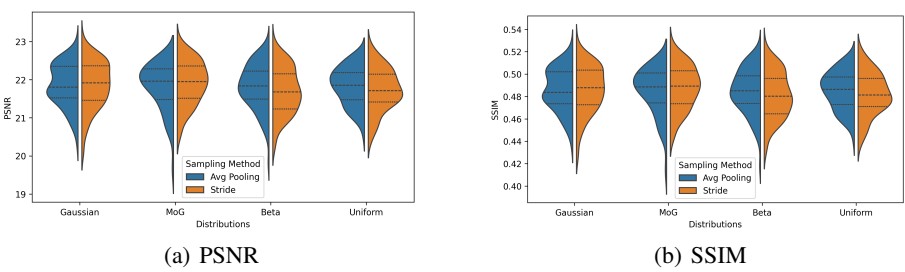

(a) PSNR  (b) SSIM

Figure 6: Violin plot of denoising results using different sampling methods. (a) PSNR results. (b) SSIM results.

 ## B   Tensor network decomposition

 ### B.1   Matrix Product Operators

A matrix product operator (MPO) (McCulloch, 2008; Hubig et al., 2017) is the TN representation of a linear operator acting on a TT format, which makes it highly efficient to handle large operators.

547 Namely, a linear operator $\boldsymbol{\mathcal{A}} : \mathbb{R}^{I_1 \times \cdots \times I_D} \to \mathbb{R}^{J_1 \times \cdots \times J_D}$. Namely, if $\boldsymbol{\mathcal{Y}} = \boldsymbol{\mathcal{A}} \boldsymbol{\mathcal{X}}$, then each entry of
548 $\boldsymbol{\mathcal{Y}}$ is given as

$$y_{\mathbf{i}} = \sum_{i_1=1}^{I_1} \cdots \sum_{i_D=1}^{I_D} \boldsymbol{A}^1_{j_1,i_1} \boldsymbol{A}^2_{j_2,i_2} \ldots \boldsymbol{A}^D_{j_D,i_D} \boldsymbol{X}^1_{i_1} \boldsymbol{X}^2_{i_2} \ldots \boldsymbol{X}^D_{i_D},$$

## B.2 Prolongation Operator

550 This work uses a specific MPO, known as the prolongation operator $\boldsymbol{\mathcal{P}}_d$ (Lubasch et al., 2018), to
551 upsample a QTT format of an image from resolution $d-1$ to $d$.

552 Consider a one-dimensional vector $\boldsymbol{x}_d \in \mathbb{R}^{2^d}$. The matrix $\boldsymbol{P}_{2^d \to 2^{d+1}}$ upsamples $\boldsymbol{x}_d$ to $\boldsymbol{x}_{d+1}$ by linear
553 interpolation between adjacent points. For example, for $d = 2$,

$$\boldsymbol{P}_{4 \to 8} = \begin{bmatrix} 1 & 0 & 0 & 0 \\ 0.5 & 0.5 & 0 & 0 \\ 0 & 1 & 0 & 0 \\ 0 & 0.5 & 0.5 & 0 \\ 0 & 0 & 1 & 0 \\ 0 & 0 & 0.5 & 0.5 \\ 0 & 0 & 0 & 1 \\ 0 & 0 & 0 & 0.5 \end{bmatrix}$$

554 The matrix $\boldsymbol{P}_{2^d \to 2^{d+1}}$ can be written as an MPO $\boldsymbol{\mathcal{P}}_{d+1}$ entry-wise

$$p_{j_1,\ldots,j_d,i_1,\ldots,i_{d+1}} = \boldsymbol{P}^1_{j_1,i_1} \ldots \boldsymbol{P}^d_{j_d,i_d} \boldsymbol{P}^{d+1}_{i_{d+1}}.$$

555 The entries are given explicitly (Lubasch et al., 2018) as

$$\boldsymbol{P}^l_{1,1}(1,1) = \boldsymbol{P}^l_{2,2}(1,1) = \boldsymbol{P}^l_{2,1}(1,2) = \boldsymbol{P}^l_{1,2}(2,2) = 1, \forall l \in [d]$$
$$\boldsymbol{P}^{d+1}_1(1) = 1, \boldsymbol{P}^{d+1}_2(1) = \boldsymbol{P}^{d+1}_2(2) = 0.5,$$

556 and other entries are zero.

557 The prolongation operator described above applies to the QTT format of one-dimensional vectors.
558 In general, this operator is the tensor product of the one-dimensional operators on each dimension:
559 $\boldsymbol{\mathcal{P}}_d^{(2)} = \boldsymbol{\mathcal{P}}_d \otimes \boldsymbol{\mathcal{P}}_d$ for 2-dimensions (images) and $\boldsymbol{\mathcal{P}}_d^{(3)} = \boldsymbol{\mathcal{P}}_d \otimes \boldsymbol{\mathcal{P}}_d \otimes \boldsymbol{\mathcal{P}}_d$ for 3-dimensions (3D
560 objects). For simplicity, since this work concerns only images, the superscript is omitted, denoting
561 the prolongation operator as $\boldsymbol{\mathcal{P}}_d$.

562 Ultimately, for a resolution $d$ image $\boldsymbol{x}_d$, and $\boldsymbol{\mathcal{X}}_d = \mathrm{Q}(\boldsymbol{x}_d)$, the upsampled image is resolution $d+1$,
563 given as $\mathrm{P}_d(\boldsymbol{x}_d) = \mathrm{Q}^{-1}(\boldsymbol{\mathcal{P}}_d \boldsymbol{\mathcal{X}}_d)$, where the linear function $\mathrm{P}_d(\cdot)$ acts on the image level.

## B.3 Recap of PuTT

565 A $2^D \times 2^D$ image, denoted as $\boldsymbol{x}_D$, can be quantized in to a $D$th order tensor $\boldsymbol{\mathcal{X}}_D = \mathrm{Q}(\boldsymbol{x}_D)$. Firstly, $\boldsymbol{x}_D$
566 is downsampled by average pooling to $\boldsymbol{x}_{D-l}$, correspondingly possesing a quantization $\boldsymbol{\mathcal{X}}_{D-l}$. Then,
567 $D-l$ QTT cores of $X_{D-l}$ can be optimized by backpropagation, returning $\boldsymbol{\mathcal{Y}}_{D-l}$. The QTT cores of
568 next resolution $\boldsymbol{\mathcal{X}}_{D-l+1}$ can be optimized similarly, initialized by the prolongation $\boldsymbol{\mathcal{P}}_{D-l+1}(\boldsymbol{y}_{D-l})$.
569 Repeat the process until the original resolution. (Loeschcke et al., 2024) demonstrates impressive
570 reconstruction capability of PuTT thanks to the QTT structure and coarse-to-fine approach. The
571 pseudocode is given in Algorithm 2.

**Algorithm 2** PuTT (Loeschcke et al., 2024)

---

**Input:** Image $\boldsymbol{x}_D$, number of iterations $T$, upsampling iterations $(t_1, \ldots, t_l)$.
**Output:** TT reconstruction $\boldsymbol{y}_D = \text{PuTT}(\boldsymbol{x}_D)$.
$d \leftarrow D - l\,, \boldsymbol{x}_d \leftarrow \text{AvgPool}(\boldsymbol{x}_D)\,, \boldsymbol{\mathcal{X}}_d \leftarrow \text{Q}(\boldsymbol{x}_d)$
**for** $t = 1 \rightarrow T$ **do**
   **if** $t \in (t_1, \ldots, t_l)$ **then**
      $d \leftarrow d + 1$
      $\boldsymbol{x}_d \leftarrow \text{AvgPool}(\boldsymbol{x}_D)$
      $\boldsymbol{\mathcal{X}}_d \leftarrow \text{Q}(\boldsymbol{x}_d)$
   **end if**
   Loss $\ell \leftarrow \text{MSE}(\boldsymbol{\mathcal{Y}}_d - \boldsymbol{\mathcal{X}}_d)$
   Update QTT cores $\boldsymbol{\mathcal{Y}}_d$ by backpropagation
**end for**
**return** $\boldsymbol{y}_D = \text{Q}^{-1}(\boldsymbol{\mathcal{Y}}_D)$

---

However, while PuTT aims to obtain better initialization by downsampling for better optimization and reconstruction, it does not account for adversarial examples or analyze the impact of downsampling on perturbations. Additionally, PuTT also minimizes the reconstruction loss on the input image, which inevitably results in the reconstruction of the perturbations. In contrast, we focus on the perturbations and propose a new optimization process introduced in the next section, aiming to reconstruct clean examples.

## C  Implementation details of adversarial attacks

We evaluate our method of defending against AutoAttack (Croce & Hein, 2020) and compare with the state-of-the-art methods as listed RobustBench benchmark (https://robustbench.github.io). For a comprehensive evaluation, we conduct experiments under all adversarial attack settings. Specifically, we set $\epsilon = 8/255$ and $\epsilon = 0.5/1.0$ for AutoAttack $l_{\text{inf}}$ and AutoAttack $l_2$ threats on CIFAR-10. On CIFAR-100, we set $\epsilon = 8/255$ for AutoAttack $l_{\text{inf}}$. On ImageNet, we set $\epsilon = 4/255$ for AutoAttack $l_{\text{inf}}$. We evaluate our method of defending against PGD+EOT (Madry et al., 2018; Athalye et al., 2018b) and present the comparisons of AT methods, AP methods, and our method. Following the guidelines of (Lee & Kim, 2023), we set $\epsilon = 8/255$ for PGD+EOT $l_{\text{inf}}$ threats on CIFAR-10, where the update iterations of PGD is 200 with 20 EOT samples.

Considering the potential robustness overestimation (Athalye et al., 2018a) caused by obfuscated gradients of purifier model, we utilize BPDA as an adaptive attack (Tramer et al., 2020; Croce et al., 2022), following the setting by (Yang et al., 2019; Lin et al., 2024a), which treats the purification step as an identity mapping during the backward pass, effectively bypassing its effect when computing gradients. In all experiments, the attacker has knowledge of both the purifier (TNP) and the classifier (Cls). The target of the attack is a new model $F$, i.e., $F(x) = Cls(TNP(x))$. The reason we chose BPDA is that the existing full gradient attacks are not applicable in TN-based AP due to the memory explosion issues associated with attacking TN optimization. In contrast to diffusion-based AP, TN is a model-free technique that does not rely on a fixed model or any parameters for gradient computation. Additionally, the iterative process in TN is a gradual optimization procedure, rather than the fixed inference iterations employed in diffusion-based methods, resulting in surrogate attacks that are difficult to apply to TN-based AP. Therefore, we empirically validated the effectiveness of our method through the existing adaptive attacks, e.g., BPDA.

Remark: Unlike conventional AP methods that rely on a specific trained model for purification, TNP is a model-free technique without any parameters or the static network architecture for gradient computation, which is an inference-time optimization strategy. Additionally, the iterative process in TNP is a dynamic, gradual optimization procedure, in contrast to the fixed-step inference in DiffPure. This dynamic nature further hinders the applicability of the gradient checkpointing technique, as there is no static computational graph or predetermined set of parameters to track and store during intermediate steps. In other words, there is no well-defined checkpoint for storing intermediate gradients, thus the gradient checkpointing technique cannot be directly applied to TNP. This is also an inherent advantage of TN-based AP, which significantly increases the difficulty of developing adaptive attacks against TNP. Our paper is the first work to introduce a model-free optimization based method. We look forward that, building on the foundation established in this work, future research

will explore adaptive attack strategies specifically tailored to TN-based AP, thereby advancing and refining the defense mechanisms of TN-based AP methods.

## D  More details of experimental settings

### D.1  Implementation details of our method

For CIFAR-10, CIFAR-100 with resolution $32 \times 32$ and ImageNet with resolution $224 \times 224$, we first upsample them into resolution $2^D \times 2^D$ image $x_D$. Based on the initial experimental results, we set $D = 8$, $l = 1$, $\alpha = 0.1$, inital $\beta = 0.008$ and $N = 1$ for the following experiments. For the scale hyperparameter $\eta$, we set $\eta = 0.1$ in all our experiments without knowing the specific attack norm. Since adversarial perturbations are very small, a fixed $\eta = 0.1$ already exceeds the scale of most attacks. Moreover, choosing a larger $\eta$ can introduce excessive noise, leading to lower-quality reconstructions. Based on our preliminary experiments, $\eta = 0.1$ offers a suitable balance and thus serves as our default setting. The table results presented in the paper are conducted under these hyperparameters. This trick creates a large enough image to downsample until the perturbations are well mixed into Gaussian noise. Furthermore, without this initial step, the semantic information can become almost indistinguishable after several downsampling steps, especially for low-resolution images. For example, if a $32 \times 32$ image is reduced with the factor of 8, the resolution $4 \times 4$ image is of poor quality. Additionally, to more clearly observe the denoising effects in visualization results, we upsample the images to resolution $D = 11$ with $\alpha = 0.05$, $\eta = 0.1$ and $N = 3$ for the experiments in Figure 4, and comparisons in different downsampled images in Figure 1. The code will be available upon acceptance, with more details provided in the configuration files.

### D.2  Implementation details of evaluation metrics

We evaluate the performance of defense methods using multiple metrics: Standard accuracy and robust accuracy (Szegedy et al., 2014) on classification tasks. For denoising tasks, we measure the Normalized Root Mean Squared Error (NRMSE, Botchkarev, 2018), Structural Similarity Index Measure (SSIM, Hore & Ziou, 2010), Peak Signal-to-Noise Ratio (PSNR) metrics between a reference image $\boldsymbol{x}$ and its reconstruction $\boldsymbol{y}$, where pixel values are in $[0, 1]$.

Normalized Root Mean Squared Error

$$\text{NRMSE}(\boldsymbol{x}, \boldsymbol{y}) = \frac{\|\boldsymbol{x} - \boldsymbol{y}\|_2}{\|\boldsymbol{x}\|_2} = \frac{\sqrt{\sum_i (\boldsymbol{x}_i - \boldsymbol{y}_i)^2}}{\sqrt{\sum_i \boldsymbol{x}_i^2}} .$$

Structural Similarity Index Measure

$$\text{SSIM}(\boldsymbol{x}, \boldsymbol{y}) = \frac{(2\mu_x \mu_y + C_1)(2\sigma_{xy} + C_2)}{(\mu_x^2 + \mu_y^2 + C_1)(\sigma_x^2 + \sigma_y^2 + C_2)},$$

where: $\mu_x$ and $\mu_y$ are the mean pixel values of images $\boldsymbol{x}$ and $\boldsymbol{y}$. $\sigma_x^2$ and $\sigma_y^2$ are the variances of $\boldsymbol{x}$ and $\boldsymbol{y}$. $\sigma_{xy}$ is the covariance between $\boldsymbol{x}$ and $\boldsymbol{y}$. $C_1$ and $C_2$ are small constants to stabilize the division.

Peak Signal-to-Noise Ratio

$$\text{PSNR}(\boldsymbol{x}, \boldsymbol{y}) = 10 \log_{10} \left( \frac{1}{\text{MSE}(\boldsymbol{x}, \boldsymbol{y})} \right) .$$

NRMSE, SSIM and PSNR evaluate reconstructed image quality from error, structural-similarity, and signal-to-noise perspectives, making them particularly suitable and comprehensive for assessing reconstruction performance. In traditional denoising and reconstruction tasks, generally a lower NRMSE, a higher SSIM, and a higher PSNR generally indicate better performance.

## E  Comparison

### E.1  Adversarial defense methods

In the development of adversarial defense methods, with the emergence of adversarial attacks, numerous methods have been proposed, including adversarial training (AT) and adversarial purification

Table 8: Comparison of defenses with vanilla model on CIFAR-10 (negative impacts are marked in red and positive impacts are marked in green). #: Using pre-trained generative model. Unseen datasets: Applying the model trained on CIFAR-10 to CIFAR-100 evaluation.

| Defense method | Clean examples | Adv. examples | Unseen attacks | Unseen datasets | Training costs | Inference costs |
|---|---|---|---|---|---|---|
| Vanilla model | ∼95% | ∼0% | ∼0% | ∼82% / ∼0% | 0 | ∼0 |
| Expectation | ≈ | ↑↑ | ↑↑ | = / ↑↑ | 0 | ∼0 |
| AT | ↓↓ | ↑↑↑ | N/A | ↓↓/↑↑↑ | ↑↑ | ∼0 |
| AP# | ↓ | ↑↑ | ↑↑ | N/A | ↑↑↑ | ↑↑ |
| TNP (ours) | ↓ | ↑↑ | ↑↑ | ↓/↑↑ | **0** | ↑↑ |

(AP). As research in this area progresses, researchers have gradually moved beyond defenses tailored to specific attacks and begun developing more general defense techniques that enhance model robustness and generalization against unseen attacks and datasets.

As mentioned before, AT predominantly consists of retraining the model on a finite set of adversarial examples, thereby conferring robustness primarily against those known perturbations. However, this process closely resembles a form of overfitting: the classifier becomes highly specialized to the attack patterns learned during training, at the expense of its performance on clean examples. As a result, standard accuracy typically degrades, and the robustness to withstand previously unseen attacks remains severely limited, as shown in Table 5.

Another class of defense methods is AP, which leverages pre-trained generative models trained on clean examples, thus can effectively defend against all types of attacks. However, AP is constrained by the specific dataset used during training, making it difficult to transfer effectively to new tasks or data distributions. As shown in Table 6, when applying the diffusion model trained on CIFAR-10 to CIFAR-100 evaluation, the standard accuracy dropped by 35.76% compared with AT.

Therefore, both mainstream defense methods face significant generalization challenges. To address this, one possible solution is to re-train the robust classifier to defend against new attacks or train a new generator on new datasets. However, such strategies incur substantial computational overhead and training costs, making them impractical for deployment in adversarial environments characterized by continuously emerging attacks, as summarized in Table 8.

To tackle these challenges with the framework of AT and AP, we propose a novel defense technique based on tensor network representation, which eliminates the need for training a powerful generative model or relying on specific dataset distributions, making it a general-purpose adversarial purification. In the experiments, TNP has shown great advantages in these challenges: 26.45% improvement in average robust accuracy over AT across different norm threats; 9.39% improvement over AP across multiple attacks; 6.47% improvement over AP across different datasets. Remarkably, TNP achieves these benefits with zero additional training cost, offering an efficient solution for adversarial purification.

## E.2 Inference time cost

Table 9: Comparison of inference time.

| Methods | CIFAR-10 | CIFAR-100 | ImageNet | Avg. |
|---|---|---|---|---|
| AT | 0.002 s | 0.002 s | 0.005 s | 0.003 s |
| DM-based AP (Nie et al., 2022) | 1.49 s | 1.50 s | 5.11 s | 2.70 s |
| AGDM (Lin et al., 2024b) | 1.73 s | 1.75 s | 5.52 s | 3.00 s |
| TNP (Ours) | 2.45 s | 2.44 s | 3.13 s | 2.67 s |

Table 9 shows the inference time of different methods on CIFAR-10, CIFAR-100, and ImageNet, which is measured on a single image. We leverage the parallelization to further improve the computa-

tional efficiency of TNP and conducted experiments on a single A5000 GPU. Specifically, AP method purifies CIFAR data at a resolution of $32 \times 32$ and ImageNet data at $256 \times 256$, whereas our method operates at a resolution of $256 \times 256$ across all datasets, which inevitably increases inference cost on CIFAR-10 and CIFAR-100. In a comparison at the same resolution of ImageNet, the diffusion-based AP method require 5.11 seconds, whereas our method only takes 3.13 seconds. Although this overhead is already lower than that of diffusion-based AP methods, it still lacks sufficient flexibility in real-world applications. We leave the study of integrating our TN-based AP technique with more advanced and faster optimization strategies for future research.

### E.3 Zero-shot adversarial defense

AT and AP methods depend heavily on external training dataset, overlooking the potential internal priors in the input itself. Among adversarial defense techniques, untrained neural networks such as deep image prior (DIP, Ulyanov et al., 2018) and masked autoencoder (MAE, He et al., 2022) have been utilized to avoid the need of extra training data (Dai et al., 2020, 2022; Lyu et al., 2023). However, although such deep learning models achieve high-quality reconstruction results, they have been shown to be susceptible to revive also the adversarial noise. This section compares two representative untrained models DIP and MAE.

Table 10: Comparison with untrained networks against AutoAttack $l_\infty$ ($\epsilon = 8/255$) on CIFAR-10.

| Defense method | Acc. | NRMSE | SSIM | PSNR |
|---|---|---|---|---|
| Clean examples | | | | |
| DIP | 90.43 | 0.0464 | 0.9565 | 32.13 |
| MAE | 88.28 | 0.0847 | 0.8842 | 26.90 |
| Ours | 82.23 | 0.0644 | 0.9203 | 29.06 |
| Adversarial examples | | | | |
| DIP | 38.28 | 0.0451 | 0.9467 | 32.53 |
| MAE | 1.56 | 0.0914 | 0.8472 | 26.24 |
| Ours | 55.27 | 0.0748 | 0.8707 | 27.77 |

Table 10 shows that although DIP and MAE have achieved remarkable standard accuracy and reconstruction quality, they deteriorate significantly under attack.

### E.4 More experiments

To ensure a fair and consistent comparison, we consider employing a robust classifier for diffusion-based AP method in Table 11.

Table 11: Standard accuracy and robust accuracy on CIFAR-10.

| Defense method | Standard Acc. | Robust Acc. |
|---|---|---|
| Strandard Training | 94.78 | 0.00 |
| Adversarial Training | 92.16 | 67.73 |
| DiffPure | 89.02 | 70.64 |
| DiffPure + AT | 90.76 | 71.68 |
| Ours + AT | 91.99 | 72.85 |

Using a robust classifier on CIFAR-10 for diffusion-based AP leads to a slight improvement in robust accuracy. Meanwhile, our method with AT consistently maintains state-of-the-art performance.

Recently, Lee & Kim (2023) conducted a thorough investigation and proposed a robust evaluation guideline using PGD+EOT. To undertake a more comprehensive evaluation, we further evaluate our method following the guidelines in this part. Table 12 shows the results on CIFAR-10, and

Table 12: Standard accuracy and robust accuracy against PGD+EOT ($l_\infty$, $\epsilon = 8/255$) on CIFAR-10.

| Type | Defense method | Standard Acc. | Robust Acc. |
|---|---|---|---|
| Adv. Training | (Pang et al., 2022) | 88.62 | 64.95 |
| | (Gowal et al., 2020) | 88.54 | 65.93 |
| | (Gowal et al., 2021) | 87.51 | 66.01 |
| DM-based AP | (Yoon et al., 2021) | 85.66 | 33.48 |
| | (Nie et al., 2022) | 91.41 | 46.84 |
| | (Lee & Kim, 2023) | 90.16 | 55.82 |
| | (Lin et al., 2024b) | 90.42 | 64.06 |
| | Ours* | 91.99 | 72.07 |

the observations are basically consistent with the existing experiments, supporting our method as a powerful defense technique and more effective than existing AT or AP methods.

# F More discussion

As we all know, the adversarial challenge of attack and defense is endless. This contradiction arises from the fundamental difference between adversarial attacks and defenses. Attacks are inherently destructive, whereas defenses are protective. This adversarial relationship places the attacker in an active position, while the defender remains passive. As a result, attackers can continually explore new attack strategies against a fixed model to degrade its predictive performance, ultimately leading to the failure of conventional defenses. The introduction of TNP has the potential to address this issue. As a model-free technique, TNP generates tensor representations solely based on the input information. These representations dynamically change with each input, preventing attackers from exploiting a fixed model to generate effective adversarial examples. This defensive mechanism allows TNP to maintain a more proactive stance in the ongoing competition between adversarial attacks and defenses.

# G Histogram, kernel density estimation results, and visualization

Figure 7 shows the histogram and kernel density estimation of adversarial perturbations on 10 images. The distribution of those perturbations progressively aligns with that of Gaussian noise as the downsampling process progresses.

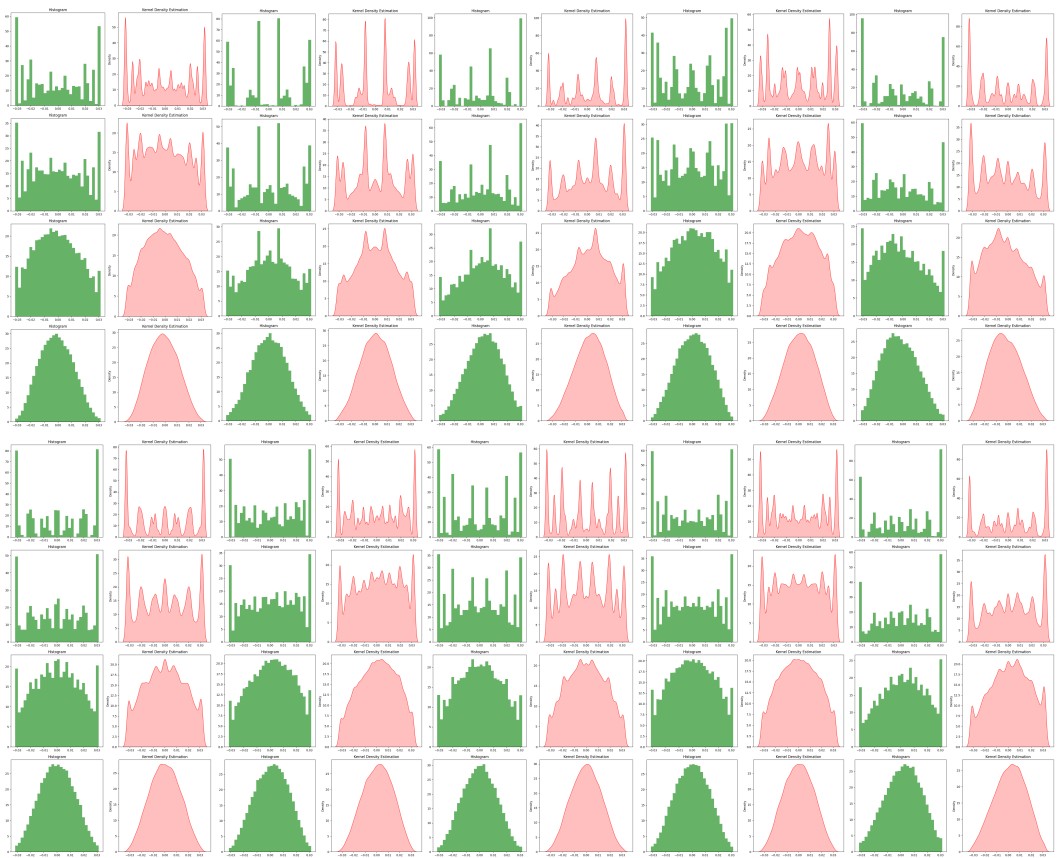

Figure 7: The histogram and kernel density estimation of adversarial perturbations in the downsampled images.

Clean examples

Adversarial examples

Reconstructed examples

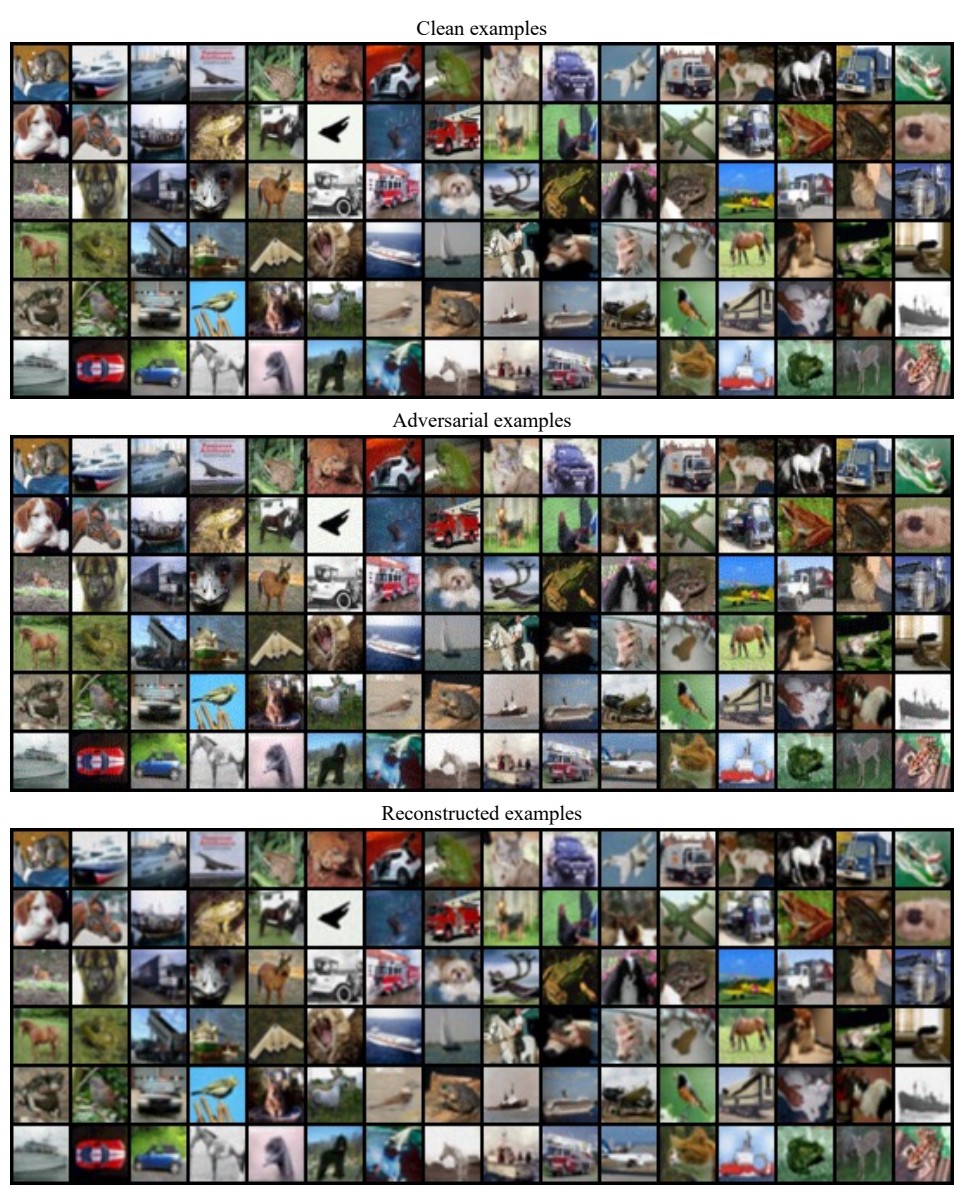

Figure 8: Clean examples (Top), adversarial examples (Middle) and reconstructed examples (Bottom) of CIFAR-10.

