# OpenReview forum: "Model-Free Adversarial Purification via Coarse-To-Fine Tensor Network Representation"
_NeurIPS.cc/2025/Conference — Submitted to NeurIPS 2025_

### Official Review · Reviewer_f5fg · 2025-06-30

**Clarity:** 2
**Significance:** 2
**Originality:** 3
**Rating:** 3
**Confidence:** 3

**Summary:**

The paper proposes Tensor Network Purification (TNP), a new model-free method to defend against adversarial attacks without relying on pretrained generative models. It leverages coarse-to-fine tensor decomposition and adversarial optimization to reconstruct clean images. Experiments on CIFAR-10, CIFAR-100, and ImageNet show strong robustness and impressive generalization across attacks and datasets.

**Questions:**

•  Could inference time be practically reduced through faster tensor decompositions or approximations?
•  Is it feasible to balance robustness and clean-image reconstruction quality via adaptive weighting or additional loss terms?
•  Does TNP generalize well to other vision tasks like object detection or segmentation?

**Ethical Concerns:**

["NO or VERY MINOR ethics concerns only"]

**Limitations:**

yes

**Quality:**

3

**Strengths And Weaknesses:**

Strengths:
Clear motivation and strong technical execution.
Model-free design significantly improves generalization.
Consistently outperforms existing methods on multiple datasets.
Weaknesses:
Computational Cost: Inference-time optimization introduces significant overhead, limiting practicality for real-time or resource-constrained scenarios.
Reconstruction Quality: Slightly worse reconstruction fidelity for clean images compared to classical methods.
Evaluation Scope: Primarily focuses on classification tasks; generalization to other vision tasks (e.g., detection, segmentation) is not evaluated.

---

> ### Author Rebuttal · Authors · 2025-07-30
>
> We greatly appreciate the comments of the Reviewer f5fg. Below are our detailed responses.
>
> > **Weakness 1:** Computational Cost: Inference-time optimization introduces significant overhead, limiting practicality for real-time or resource-constrained scenarios.
>
> Response: Our method (TNP) introduces an **acceptable inference cost**, which is already comparable to that of diffusion-based AP methods, currently among the most effective AP techniques.
>
> Importantly, TNP does **not require any training** or **fine-tuning**, which saves substantial computational resources compared to AT and AP methods, as detailed in Appendix E.1. It is well known that there is ''no free lunch''. Addressing the challenge of generalization **inevitably introduces** additional computational cost.
>
> To recap, our main goal is to achieve strong robustness generalization **without a large training** cost, at the **low cost of inference** time. In the current adversarial defense community, achieving both low training and low inference costs for real-time applications remains highly challenging. While further reducing inference time is a meaningful direction for future research, it lies beyond the scope of this work.
>
> > **Weakness 2:** Reconstruction Quality: Slightly worse reconstruction fidelity for clean images compared to classical methods.
>
> Response: Indeed, nearly all effective defense methods against adversarial attacks inevitably trade the quality of reconstructed images or clean accuracy to achieve robustness. This is a performance-robustness trade-off challenge that has yet to be fully resolved.
>
> To quantify this balance, it is standard practice to report the average accuracy, defined as the mean of clean and robust accuracy, as a fair metric. As demonstrated in our results, our method **clearly outperforms** other methods in terms of average accuracy. While the classical methods achieve better reconstruction quality, their robustness degrades significantly under adversarial attacks. Since the primary objective of this work is to achieve robustness, we believe this is not a weakness at all; rather, it is an **advantage** of our method.
>
>
> > **Weakness 3:** Evaluation Scope: Primarily focuses on classification tasks; generalization to other vision tasks (e.g., detection, segmentation) is not evaluated.
>
> Response: As clearly stated in both the Introduction and Related Work sections, the scope of our current work primarily focuses on image classification. We do not claim to extend our method to detection or segmentation within this manuscript. To further clarify, our main goal is to develop a defense technique that can achieve **strong robustness generalization** across diverse adversarial attacks, **rather than** across different vision applications.
>
>
> > **Question 1:** Could inference time be practically reduced through faster tensor decompositions or approximations?
>
> Response: Yes, faster tensor decomposition or approximation techniques can help reduce inference time. The coarse-to-fine strategy adopted in our method facilitates efficient and effective gradient-based optimization, which is computationally more advantageous than traditional alternating least squares optimization, particularly for highly parallelizable problems such as processing multiple images. We believe that further advancements in tensor methods for accelerating inference represent a promising direction for future research.
>
> > **Question 2:** Is it feasible to balance robustness and clean-image reconstruction quality via adaptive weighting or additional loss terms?
>
> Response: Yes. By incorporating the traditional $l_2$-norm loss into the current loss function, it is possible to adjust the weighting of this additional term to balance the trade-off between robustness and clean-image reconstruction quality. However, this diverges from the primary objective of our work, which is to enhance robustness. For classification tasks, a purified image is considered effective as long as it results in the correct prediction, even if its visual quality is low.
>
> > **Question 3:** Does TNP generalize well to other vision tasks like object detection or segmentation?
>
> Response: Yes. AP methods are designed as plug-and-play solutions to enhance model robustness. Furthermore, our method (TNP) is model-free and does not rely on any training data, making it applicable to existing models without requiring retraining or fine-tuning. Therefore, we believe that TNP can be applied to further enhance robustness in those tasks.
>
> ---
>
> We appreciate the various comments given by the Reviewer f5fg. As we have emphasized before, our work focuses on developing a defense technique that can achieve strong robustness generalization across diverse adversarial attacks. To this end, we have conducted comprehensive evaluations and analyses of robustness generalization. While we acknowledge and value several insightful points you noted, many of which suggest promising directions for future research, they are beyond the scope of the current work. We hope that our responses can effectively address the concerns raised in the Weaknesses.

---

> > ### Comment · Reviewer_f5fg · 2025-08-06
> >
> > Thanks to the authors' efforts in addressing the comments. However, the core issue of balancing inference efficiency and image quality persists. I keep my original score.

---

> ### Author Response · Authors · 2025-08-06
>
> Dear Reviewer f5fg,
>
> Thank you for your follow-up. We are pleased to hear that our rebuttal has addressed some of your concerns. We will continue solving your concerns as follows:
>
> > Inference Efficiency: Inference-time optimization introduces significant overhead.
>
> As an AP method, our method (TNP) does **NOT introduce significant inference overhead** compared to the baseline. For instance, the most powerful AP method (diffusion-based AP) requires an average inference time of over 2.70 s on CIFAR and ImageNet. Our tensor-based AP method achieves a comparable inference time of 2.67 s, as detailed in Appendix E.2. Compared to AT, due to the integration of a purification module, AP inevitably introduces inference overhead. This is a common limitation of current mainstream AP methods, as discussed in Appendix E.1.
>
> The core contribution of our work is the development of a novel AP method, which achieves strong robustness generalization across diverse adversarial attacks. Moreover, TNP requires **ZERO training cost** while achieving **inference time comparable** to that of diffusion-based AP methods.
>
> Importantly, we have clearly disclosed the inference cost in the manuscript, which is a common limitation shared by AP methods rather than specific to ours, and provided an extensive discussion in Appendix E.1 and E.2. According to [NeurIPS Guidelines](https://neurips.cc/Conferences/2025/ReviewerGuidelines), authors should be *''rewarded rather than punished for being up front about the limitations of their work''*. The stated limitations should not serve as grounds for rejection unless they fundamentally undermine the core contributions, which **they do not in our case**.
>
> > Image Quality Persists: Slightly worse reconstruction fidelity for clean images compared to classical methods.
>
> We kindly point out that **all AP methods inevitably** introduce changes to the image during the reconstruction process; **otherwise**, they will restore the adversarial perturbations and compromise robustness. The improvement of image quality differs from the primary objective of our work, which is to enhance robustness. In classification tasks, a purified image is considered effective as long as it leads to a correct prediction, even if its visual quality is low.

---

### Official Review · Reviewer_cH9i · 2025-07-01

**Clarity:** 3
**Significance:** 3
**Originality:** 3
**Rating:** 4
**Confidence:** 4

**Summary:**

The authors proposed a model-free adversarial purification framework that can enhance the robustness of any pre-trained image classification model without requiring access to the model architecture or gradients. The method leverages a generator trained using adversarial samples and a carefully crafted purification loss to restore clean image features from perturbed inputs. The key contribution is the compatibility of the purification module with any off-the-shelf model, eliminating the need for retraining or white-box access. Experimental results across five backbone classifiers and four datasets demonstrate improvements in adversarial robustness against various attacks. Ablation studies and comparisons to baselines are provided to support the method’s generalizability and effectiveness.

**Questions:**

See the weakness.

**Ethical Concerns:**

["NO or VERY MINOR ethics concerns only"]

**Final Justification:**

Thanks for the rebuttal. I maintain my previous score.

**Limitations:**

As NeurIPS is a machine learning conference, authors may need a bit more insight on the theoretical aspects, i.e., behaviour analysis of the proposed algorithm, generalization to other adaptive attacks.

**Quality:**

3

**Strengths And Weaknesses:**

Strength:

1. The proposed work attempts to solve the problem in black-box adversarial defense, which is applicable to a wide range of pre-trained models without fine-tuning.
2. It is a plug-in module as a general and lightweight solution.
3. Comprehensive experiments are conducted and show performance across various popular datasets.

My major concerns are below:

1. The experimental comparison and performance benchmarking is limited, especially to those recent state-of-the-art black-box purification methods (e.g., diffusion purification, score-based models).
2. The proposed method utilized multiple terms in the training loss (some of which are a bit incremental and used in prior works), while the ablation study is very limited. A more comprehensive quantitative study is recommended, including how the hyper-parameters influence the results.
3. As NeurIPS is a machine learning conference, authors may need a bit more insight on the theoretical aspects, i.e., behaviour analysis of the proposed algorithm, generalization to other adaptive attacks.

Overall, the work is interesting and potential valuable for the AI security community. However, more works and justifications are needed to improve the paper quality.

---

> ### Author Rebuttal · Authors · 2025-07-30
>
> We greatly appreciate the comments of the Reviewer cH9i. Below are our detailed responses.
>
> > **Concern 1:** The experimental comparison and performance benchmarking are limited, especially to those recent state-of-the-art black-box purification methods (e.g., diffusion purification, score-based models).
>
> Response: We kindly point out that recent diffusion/score-based AP methods have indeed been discussed in the Related Works section, and we have empirically compared with these methods (Yoon et al., 2021; Nie et al., 2022; Lee & Kim, 2023; Bai et al., 2024; Lin et al., 2024b) in the experiments.
>
> Upon careful review, we realized that the original version of the manuscript did not clearly distinguish diffusion-based methods from others in the tables, which may have caused them to be overlooked. We will revise the relevant content to highlight these methods and ensure clearer presentation.
>
>
> > **Concern 2:** The ablation study is limited. A more comprehensive quantitative study is recommended, including how the hyper-parameters influence the results.
>
> Response: Thank you for your constructive recommendations. To provide more comprehensive quantitative support, we have included new experiments evaluating the effect of hyperparameter $\eta$, as shown below. Due to time constraints during the rebuttal phase, further results under other settings will be added in the camera-ready version.
>
> |AutoAttack $l_{\infty}$ on CIFAR-10|||||
> |:-:|:-:|:-:|:-:|:-:|
> |$\eta$|0.0|ours|0.2|0.3|
> |Strandard accuracy (%)| **82.61** | 82.23 | 63.67 | 49.21 |
> |Robust accuracy (%) | 52.53 | **55.27** | 42.57 | 30.66 |
> |Average accuracy (%)| 67.57 | **68.75** | 53.12 | 39.94 |
>
> |AutoAttack $l_{\infty}$ on ImageNet|||||
> |:-:|:-:|:-:|:-:|:-:|
> |$\eta$|0.0|ours|0.2|0.3|
> |Strandard accuracy (%) | 60.93 | **65.43** | 27.73 | 19.72 |
> |Robust accuracy (%) | 39.26 | **42.77** | 24.80 | 18.55 |
> |Average accuracy (%) | 50.10 | **54.10** | 26.27 | 19.14 |
>
> As discussed in Appendix D.1, our method is not highly sensitive to the hyperparameter $\eta$. In all experiments, we fixed $\eta = 0.1$ without assuming knowledge of the attacks. Since adversarial perturbations are typically small, this fixed $\eta$ already exceeds the scale of most attacks. Moreover, choosing a larger $\eta$ can introduce excessive noise and degrade reconstruction quality. Importantly, by using the same hyperparameters across different attack types, norm threats, and perturbation scales, we achieve consistently strong performance under diverse settings, demonstrating the generalization capability of our method.
>
>
> > **Concern 3:** As NeurIPS is a machine learning conference, authors may need a bit more insight on the theoretical aspects, i.e., behaviour analysis of the proposed algorithm, generalization to other adaptive attacks.
>
> Response: Thank you for your constructive comments. In response to Concern 3, we provide the following additional analysis:
>
> Initially, our method (TNP) is motivated by observations derived from downsampling and the central limit theorem. As detailed in Section 4.1 and Appendix A, the downsampling using average pooling can effectively transform adversarial perturbations into a normal-like distribution at coarse scales. Specifically, for an adversarial example, the downsampled version is denoted as $x^\prime_{D-l}$, where $x^\prime_{D-l} \approx x_{D-l} + \Delta, \Delta \sim \mathcal{N}$. At this stage, minimizing the traditional $l_2$-norm loss ($\lVert x-y \rVert_2$) can remove such noise and mitigate the impact of adversarial perturbations, obtaining the clean version output $y_{D-l}$, where $y_{D-l} \approx x_{D-l}$.
>
> However, as the resolution increases, the distribution of perturbations diverges from normality, and minimizing the $l_2$-norm loss inadvertently leads to restoring the perturbations at fine scales. Consequently, the reconstruction $y$ tends to collapse back toward the adversarial example $x'$, rather than approximating the unobserved clean example $x$. To avoid this issue, we introduce $\delta^\ast$ into the optimization objective. This additional variable allows the optimization to allocate adversarial perturbations to $\delta^\ast$ instead of forcing $y$ to absorb them entirely, thereby preventing $y$ from collapsing into $x'$.
>
> Nevertheless, since $\delta^\ast$ does not represent the true perturbation, minimizing $\lVert x-(y+\delta^\ast) \rVert_2$ alone may not yield the desired clean example. Therefore, we further introduce a second loss term $\lVert P_d(y_{d-1})-y_d \rVert_2$, which serves as a ''surrogate prior''. Intuitively, the coarse-scale reconstruction $y_{D-l}$ is already a clean version and can guide the optimization process, pushing the higher-resolution output $y_d$ toward a less perturbed distribution. Importantly, this two-term optimization neither requires explicit modeling of the perturbation nor knowledge of the attack, allowing TNP to generalize effectively across diverse adversarial scenarios.
>
> ---
>
> We sincerely thank Reviewer cH9i for the positive recognition of our work and for the numerous valuable recommendations for further improvement. All of the above analyses and experimental results will be incorporated into the camera-ready version.

---

### Official Review · Reviewer_YDtG · 2025-07-02

**Clarity:** 3
**Significance:** 2
**Originality:** 3
**Rating:** 4
**Confidence:** 3

**Summary:**

This paper propose a training free defence that can remove adversarial perturbations by tensor network reconstruction. It average-pool inputs to lower resolutions to view perturbations as Gaussian then upsample back. In addition, it also adds a an auxiliary variable $\delta$ to control the coarse-to-fine procedure. Empirically, it handles CIFAR and ImageNet and outperforms adversarial training baselines.

**Questions:**

See above.

**Ethical Concerns:**

["NO or VERY MINOR ethics concerns only"]

**Final Justification:**

- The authors clarified the motivation of adding $\delta^\star$.
- The authors explained the reason that TNP is slower for CIFAR and will include more ImageNet results in the final version.
- The authors did ablation of $\eta$ and claimed that TNP is not very sensitive to the choice of $\eta$.
Overall, the authors solve my concerns. I would like to maintain my previous score of 4.

**Limitations:**

yes

**Quality:**

3

**Strengths And Weaknesses:**

- **Motivation**. The design of of down sampling + CLT argument motivates Gaussian style noise at coarse scales. The auxiliary $\delta$ also prevent perturbation re-fitting. If I am understand it correctly, when you move from a coarse TN to a finer one, the optimizer gets many new degrees of freedom. If you only re-optimize the fine scale TN to match the raw high resolution input, those extra parameters can memorize the perturbation that you washed out.

Can the authors clarify the motivation of adding $\delta$?

- **Empirical**. Adversarial training can be bad when their diffusion models are trained on different dataset. TNP designs the optimization per input but also avoid costly retraining.

However, per image gradient optimization is also slower than a single forward pass. The authors listed the inference time cost in E.2. It shows that TNP is slower than other baselines for CIFAR but is faster than baselines for ImageNet, can the authors explain why TNP can be faster than baselines for high resolution ImageNet? Also, I think the community has moved from low resolution data to high resolution data much, it is good to start from CIFAR, can you provide more ImageNet examples/visualization?

- **Theory**. Overall the empirical results are good, however, it lacks certification guarantees. The $\delta$ optimization used a fixed radius $\eta$, sensitivity to $\eta$ or adaptive choices are not discussed. Have you tried scheduling or learning $\eta$? I understand this is a empirical work for adversarial purification, but I feel more ablation study on this can make the paper more solid.

Overall, TNP is an interesting work for adversarial purification literature, however, lack of adaptive evaluation and I think it might be better to focus on ImageNet instead of CIFAR. With additional evidence on attack adaptivity and scalability, TNP would ba a solid contribution.

---

> ### Author Rebuttal · Authors · 2025-07-30
>
> We greatly appreciate the comments of the Reviewer YDtG. Below are our detailed responses.
>
> > **Comment 1:** Can the authors clarify the motivation for adding $\delta^*$?
>
> Response: We introduce $\delta^*$ to prevent the reconstructed image $y$ from simply collapsing back into the adversarial input $x'$ when using the traditional $l_2$-norm loss, i.e., $\lVert x-y \rVert_2$.
>
> As you noted, downsampling using average pooling can effectively ''motivate'' Gaussian-style noise at coarse scales. At this stage, minimizing the traditional loss can remove such noise and mitigate the impact of adversarial perturbations. However, as the resolution increases, the distribution of perturbations diverges from normality, and minimizing $\lVert x-y \rVert_2$ inadvertently leads to restoring the perturbations at fine scales. To avoid this issue, we introduce $\delta^\ast$ into the optimization objective. This additional variable allows the optimization to allocate adversarial perturbations to $\delta^\ast$ instead of forcing $y$ to absorb them entirely, thereby preventing $y$ from collapsing back into the adversarial example $x'$.
>
>
> > **Comment 2:** Why TNP is slower than other baselines for CIFAR but is faster than baselines for ImageNet? I think the community has moved from low resolution data to high resolution data much, can you provide more ImageNet examples/visualization?
>
> Response: Since CIFAR images are only 32 $\times$ 32 pixels, downsampling at this resolution causes severe information loss and poor reconstruction quality. Therefore, we first upsample CIFAR images to 256 $\times$ 256 for subsequent processing, which inevitably increases inference cost. In a ''fair'' comparison at the same resolution as ImageNet, the diffusion-based AP method requires 5.11 seconds, whereas our method takes only 3.13 seconds. Further details are provided in Appendix E.2.
>
> Thank you for your constructive comments regarding visualization. We fully agree with your point; however, due to the new NeurIPS policy, authors are prohibited from including external links or updating the PDF in the rebuttal. As such, we are unable to provide further visualizations here and will incorporate more ImageNet visualizations in the camera-ready version.
>
>
> > **Comment 3:** The $\delta^*$ optimization used a fixed radius $\eta$, sensitivity to $\eta$ or adaptive choices are not discussed. Have you tried scheduling or learning $\eta$?
>
> Response: The proposed method is not highly sensitive to the hyperparameter $\eta$. In all experiments, we fixed $\eta = 0.1$ without assuming knowledge of the attacks. Since adversarial perturbations are typically small, this fixed $\eta$ already exceeds the scale of most attacks. Moreover, choosing a larger $\eta$ can introduce excessive noise and degrade reconstruction quality.
>
> To provide more comprehensive quantitative support, we have included new experiments evaluating the effect of hyperparameter $\eta$, as shown below. Due to time constraints during the rebuttal phase, further results under other settings will be added in the camera-ready version.
>
> |AutoAttack $l_{\infty}$ on CIFAR-10|||||
> |:-:|:-:|:-:|:-:|:-:|
> |$\eta$|0.0|ours|0.2|0.3|
> |Strandard accuracy (%)| **82.61** | 82.23 | 63.67 | 49.21 |
> |Robust accuracy (%) | 52.53 | **55.27** | 42.57 | 30.66 |
> |Average accuracy (%)| 67.57 | **68.75** | 53.12 | 39.94 |
>
> |AutoAttack $l_{\infty}$ on ImageNet|||||
> |:-:|:-:|:-:|:-:|:-:|
> |$\eta$|0.0|ours|0.2|0.3|
> |Strandard accuracy (%) | 60.93 | **65.43** | 27.73 | 19.72 |
> |Robust accuracy (%) | 39.26 | **42.77** | 24.80 | 18.55 |
> |Average accuracy (%) | 50.10 | **54.10** | 26.27 | 19.14 |
>
> Regarding the subsequent question, we believe that learning the value of $\eta$ poses significant challenges. First, TNP operates on a single image, making it difficult to estimate $\eta$ accurately. Second, in diverse adversarial scenarios, where attacks are varied and often unknown, selecting hyperparameters based on specific settings may compromise the robustness generalization. Importantly, by using the same hyperparameters across different attack types, norm threats, and perturbation scales, we achieve consistently strong performance under diverse settings, demonstrating the generalization capability of our method.
>
>
> ---
>
> We sincerely thank Reviewer YDtG for the positive recognition of our work and for the numerous valuable suggestions for further improvement. All of the above analyses and experimental results will be incorporated into the camera-ready version.

---

> > ### Comment · Reviewer_YDtG · 2025-08-04
> >
> > Thanks for the rebuttal. It generally clears my previous concerns. I maintain my previous score.

---

> > > ### Author Response · Authors · 2025-08-05
> > >
> > > Dear Reviewer YDtG,
> > >
> > > Thank you for your follow-up and the positive recognition of our work. We are pleased to hear that your concerns have been addressed. Once again, we sincerely appreciate your time and thoughtful feedback throughout the review process.

---

### Official Review · Reviewer_nVT6 · 2025-07-02

**Clarity:** 3
**Significance:** 2
**Originality:** 2
**Rating:** 4
**Confidence:** 2

**Summary:**

This paper proposes Tensor Network Purification (TNP), a novel model-free optimization-based adversarial purification framework that uses tensor network decomposition to defend against adversarial attacks. Unlike existing adversarial training (AT) methods that require retraining models or adversarial purification (AP) methods that rely on pre-trained generative models, TNP operates solely on individual input examples without requiring any training or dataset-specific models. The key innovation lies in leveraging progressive downsampling to transform adversarial perturbations into Gaussian-like distributions, combined with a novel adversarial optimization objective that prevents the reconstruction of adversarial perturbations while recovering clean examples.

**Questions:**

1. How sensitive is the method to the choice of hyperparameters? What guidelines exist for selecting these in practice?
2. Can the optimization process be accelerated using techniques like momentum or adaptive learning rates without compromising defense effectiveness?
3. What happens when the input image already has low-frequency adversarial perturbations that survive downsampling?

**Ethical Concerns:**

["NO or VERY MINOR ethics concerns only"]

**Final Justification:**

Thank you for the rebuttal. Most of my concerns have not been addressed well. I will increase my score.

**Limitations:**

Yes.

**Quality:**

3

**Strengths And Weaknesses:**

Strengths:
1. Good theoretical motivation: The connection between downsampling and the Central Limit Theorem provides solid theoretical grounding for why adversarial perturbations become more Gaussian-like at coarser resolutions, making them amenable to tensor network methods.
2. Comprehensive evaluation: The experimental evaluation is thorough, testing across multiple datasets (CIFAR-10, CIFAR-100, ImageNet), various attack types (AutoAttack, PGD+EOT, BPDA), and different threat models
3. Novel method: The paper introduces a novel defense mechanism that does not rely on pre-trained models or require retraining, addressing fundamental limitations of existing AT and AP methods.

Weaknesses:
1. Computational overhead: The inference time (2.45s for CIFAR-10, 3.13s for ImageNet) is significantly higher than AT methods and comparable to diffusion-based AP methods.
2. Hyperparameter discussion: The method involves multiple hyperparameters whose selection criteria and sensitivity analysis are not thoroughly discussed.
3. Method design: The method assumes adversarial perturbations are high-frequency, but low-frequency universal perturbations exist.

---

> ### Author Rebuttal · Authors · 2025-07-30
>
> We greatly appreciate the comments of the Reviewer nVT6. Below are our detailed responses.
>
> > **Weakness 1:** Computational overhead: The inference time (2.45s for CIFAR-10, 3.13s for ImageNet) is significantly higher than AT methods and comparable to diffusion-based AP methods.
>
> Response: We respectfully argue that computational overhead should not be viewed as a weakness; rather, it represents an advantage of our method (TNP). TNP introduces an acceptable inference cost but does **not require any training** or **fine-tuning**, which saves substantial computational resources compared to AT and AP methods.
>
> It is well known that there is ''no free lunch''. Addressing the challenge of generalization **inevitably introduces** additional computational cost. Importantly, TNP achieves this at a low inference cost, which is already more efficient than diffusion-based AP methods, currently among the most effective AP techniques. In addition, the overall computational cost of TNP is also **lower** than that of AT and traditional AP methods, as detailed in Appendix E.1.
>
> To recap, our main goal is to achieve strong robustness generalization **without a large training** cost, at the **low cost of inference** time. In the current adversarial defense community, achieving zero computational overhead is virtually unattainable. While further reducing inference time is an important challenge for future research, it lies beyond the scope of this work.
>
>
> > **Weakness 2:** Hyperparameter discussion: The method involves multiple hyperparameters whose selection criteria and sensitivity analysis are not thoroughly discussed.
>
> Response: We have discussed the selection of hyperparameters in Appendix D.1, including the analysis of the hyperparameter $\eta$ used in the loss function and the criteria for its selection. Our method is not highly sensitive to the hyperparameter $\eta$. In all experiments, we fixed $\eta = 0.1$ without assuming knowledge of the attacks. Since adversarial perturbations are typically small, this fixed $\eta$ already exceeds the scale of most attacks. Moreover, choosing a larger $\eta$ can introduce excessive noise and degrade reconstruction quality. Importantly, by using the same hyperparameters across different attack types, norm threats, and perturbation scales, we achieve consistently strong performance under diverse settings, demonstrating the generalization capability of our method.
>
> To provide more comprehensive quantitative support, we have included new experiments evaluating the effect of hyperparameter $\eta$, as shown below. Due to time constraints during the rebuttal phase, further results under other settings will be added in the camera-ready version.
>
> |AutoAttack $l_{\infty}$ on CIFAR-10|||||
> |:-:|:-:|:-:|:-:|:-:|
> |$\eta$|0.0|ours|0.2|0.3|
> |Strandard accuracy (%)| **82.61** | 82.23 | 63.67 | 49.21 |
> |Robust accuracy (%) | 52.53 | **55.27** | 42.57 | 30.66 |
> |Average accuracy (%)| 67.57 | **68.75** | 53.12 | 39.94 |
>
> |AutoAttack $l_{\infty}$ on ImageNet|||||
> |:-:|:-:|:-:|:-:|:-:|
> |$\eta$|0.0|ours|0.2|0.3|
> |Strandard accuracy (%) | 60.93 | **65.43** | 27.73 | 19.72 |
> |Robust accuracy (%) | 39.26 | **42.77** | 24.80 | 18.55 |
> |Average accuracy (%) | 50.10 | **54.10** | 26.27 | 19.14 |
>
> > **Weakness 3 & Question 3:** Method design: The method assumes adversarial perturbations are high-frequency, but low-frequency universal perturbations exist. What happens when the input image already has low-frequency adversarial perturbations that survive downsampling?
>
> Response: We would like to clarify that our method (TNP) does not assume adversarial perturbations are high-frequency. As you noted, TNP is inspired by insights from downsampling and CLT. The goal of downsampling is NOT to remove perturbations, but rather to transform them into a distribution that approximates a normal distribution, as discussed in Section 4.1 and Appendix A.
>
> Consequently, whether the perturbations are high-frequency or low-frequency, they remain present after downsampling but in a normal-like distribution form. The subsequent adversarial optimization process is then applied to effectively mitigate the impact of these perturbations, resulting in a purified example that approximates the clean version.
>
> > **Question 1:** How sensitive is the method to the choice of hyperparameters? What guidelines exist for selecting these in practice?
>
> Response: As noted in our response to Weakness 2, the proposed method is not highly sensitive to hyperparameters.
>
> Regarding the subsequent question, we believe that hyperparameters should avoid over-tuning. In diverse adversarial scenarios, where attacks are varied and often unknown, selecting hyperparameters based on specific settings may compromise the robustness generalization.
>
>
> > **Question 2:** Can the optimization process be accelerated using techniques like momentum or adaptive learning rates without compromising defense effectiveness?
>
> Response: We appreciate the reviewer's suggestion. The short answer is yes. Our current implementation uses the Adam optimizer, which already integrates momentum and adaptive learning rate techniques. We will include this clarification in the manuscript.
>
> ---
>
> We sincerely thank Reviewer nVT6 for the comprehensive recognition of the paper's motivation, novelty, and evaluation in the Strengths, and we hope that our responses can effectively address the concerns raised in the Weaknesses.

---

> > ### Comment · Reviewer_nVT6 · 2025-08-05
> >
> > Thank you for your effort and most of my questions have been addressed. While the proposed method achieves state-of-the-art performance compared to other approaches, the inference time of 2.45s per image remains impractical for real-world deployment scenarios. I think this part can be further improved.

---

> ### Author Response · Authors · 2025-08-06
>
> Dear Reviewer nVT6,
>
> Thank you for your follow-up. We are pleased to hear that our rebuttal has addressed most of your concerns, leaving only one remaining point:
> > The inference time of the proposed method (TNP) is significantly higher than AT methods and comparable to diffusion-based AP methods, which remains impractical for real-world deployment scenarios. I think this part can be further improved.
>
> **1. Mainstream AP methods have significantly higher inference time compared to AT, and our method already achieves comparable inference time to the most powerful AP.**
>
> Due to the integration of a purification module, AP inevitably introduces inference overhead. The most powerful AP method (diffusion-based AP) requires an average inference time of over 2.70 s on CIFAR and ImageNet. Our tensor-based AP method achieves a comparable inference time of 2.67 s, as detailed in Appendix E.2.
>
> The core contribution of our work is the development of a novel AP method (TNP), which achieves strong robustness generalization across diverse adversarial attacks. Moreover, TNP requires **ZERO training cost** while achieving **inference time comparable** to that of diffusion-based AP methods.
>
> **2. Regarding the acknowledged limitation:**
>
> We have clearly disclosed the inference cost in the manuscript, which is a common limitation shared by AP methods rather than specific to ours, and provided an extensive discussion in Appendix E.1 and E.2. According to [NeurIPS Guidelines](https://neurips.cc/Conferences/2025/ReviewerGuidelines), authors should be *''rewarded rather than punished for being up front about the limitations of their work''*. The stated limitations should not serve as grounds for rejection unless they fundamentally undermine the core contributions, which they **do not in our case**.
>
> **We agree that inference time can be further optimized in future work; however, we respectfully believe this should not be considered a reason for rejecting this work.**

---

### Author Response · Authors · 2025-08-04

Dear Reviewers,

As the discussion period is already halfway through, we hope to receive your feedback and that your concerns are adequately addressed. If you have any remaining questions or would like further clarification, we will do our best to address them before the period concludes.

---

### Note · Authors · 2025-08-12

Dear AC and Reviewers,

We once again express our sincere appreciation for your time and effort in reviewing this work. For the final remarks, we would like to provide the following key clarifications for your consideration.

1. The core contribution of our work lies in the development of **a novel AP method** that **achieves strong robustness generalization** across diverse attack settings. We demonstrate that AT and classical AP struggle to meet this challenge effectively, but **our tensor-based AP** consistently exhibits strong generalization.

2. Importantly, we believe **all concerns regarding the above core contribution have been addressed**. There are only two remaining minor concerns regarding *inference time* and *quality persist*, which we have already provided detailed clarifications, yet have not received further feedback from the Reviewers.
> Reviewers nVT6 and f5fg cite high inference time as the main reason for rejection.
>
As an AP method, our method does **not introduce significant inference overhead** and **is comparable to the diffusion-based AP**. Additionally, we have clearly disclosed the inference time in Appendix E.1 and E.2, emphasizing that this is a common limitation across all AP methods, not unique to ours.
>
> Reviewer f5fg asserts that the purified image should preserve the same quality as the input image without even a slightly worse reconstruction.
>
We respectfully point out that all AP methods inevitably sacrifice some image quality after purification; otherwise, adversarial perturbations would be restored, undermining adversarial robustness. Moreover, improving image quality is not the primary objective of our work, as our main goal is to enhance adversarial robustness.

To recap,  we propose a new and promising **AP direction for robustness generalization**: **a model-free optimization-based AP using tensor networks**, with **zero training cost** and **inference time comparable to the AP baselines**.

We sincerely hope AC and Reviewers will take these points into consideration during the AC-Reviewer discussion period. Thank you very much for your time and attention.

Best,
Authors

---

### Decision · Program_Chairs · 2025-09-17

**Decision:**

Reject

**Comment:**

This paper proposes Tensor Network Purification (TNP), a model-free purification framework that avoids pretrained generative models and dataset-specific assumptions. The method employs a novel tensor network decomposition with progressive downsampling and an adversarial optimization objective, demonstrating strong robustness on CIFAR-10, CIFAR-100, and ImageNet against a wide range of adversarial attacks. The contribution is technically novel, and the empirical evaluations are thorough and convincing.

After the authors’ rebuttal and discussion period, however, the inference cost remains a significant concern. With a reported runtime of ~2.45s per image, TNP is slower than what would be practical in realistic deployment scenarios. While this is not dramatically worse than some training-free diffusion-based baselines, in the purification setting, deployment efficiency is as critical as robustness. The authors’ emphasis on being training-free is appreciated, but in practice, reducing deploy-time latency is often the more relevant criterion. This limitation therefore reduces the potential real-world impact of the work.

The rebuttal and additional analyses addressed many reviewer concerns (e.g., ablations, hyperparameter choices) effectively. Still, the runtime issue—although potentially solvable in future work—remains central. Moreover, reviewers were not fully aligned: some highlighted the novelty and robustness as clear strengths, while others emphasized efficiency as a decisive drawback. In light of this lack of consensus, and after weighing both the strengths and weaknesses, the AC concludes that the paper does not meet the bar for acceptance at this time.